# Genome sequence and analysis of the Japanese morning glory *Ipomoea nil*

Atsushi Hoshino[1,2,*], Vasanthan Jayakumar[3,*], Eiji Nitasaka[4], Atsushi Toyoda[5], Hideki Noguchi[5], Takehiko Itoh[6], Tadasu Shin-I[5], Yohei Minakuchi[5], Yuki Koda[3], Atsushi J. Nagano[7,8], Masaki Yasugi[7,†], Mie N. Honjo[7], Hiroshi Kudoh[7], Motoaki Seki[9,10], Asako Kamiya[9], Toshiyuki Shiraki[11], Piero Carninci[12], Erika Asamizu[13,†], Hiroyo Nishide[1], Sachiko Tanaka[1], Kyeung-Il Park[1,14], Yasumasa Morita[1,†], Kohei Yokoyama[4], Ikuo Uchiyama[1,2], Yoshikazu Tanaka[15], Satoshi Tabata[13], Kazuo Shinozaki[9], Yoshihide Hayashizaki[16], Yuji Kohara[5], Yutaka Suzuki[17], Sumio Sugano[18], Asao Fujiyama[5,19], Shigeru Iida[1,2,†] & Yasubumi Sakakibara[3]

*Ipomoea* is the largest genus in the family Convolvulaceae. *Ipomoea nil* (Japanese morning glory) has been utilized as a model plant to study the genetic basis of floricultural traits, with over 1,500 mutant lines. In the present study, we have utilized second- and third-generation-sequencing platforms, and have reported a draft genome of *I. nil* with a scaffold N50 of 2.88 Mb (contig N50 of 1.87 Mb), covering 98% of the 750 Mb genome. Scaffolds covering 91.42% of the assembly are anchored to 15 pseudo-chromosomes. The draft genome has enabled the identification and cataloguing of the *Tpn1* family transposons, known as the major mutagen of *I. nil*, and analysing the dwarf gene, *CONTRACTED*, located on the genetic map published in 1956. Comparative genomics has suggested that a whole genome duplication in Convolvulaceae, distinct from the recent Solanaceae event, has occurred after the divergence of the two sister families.

[1] National Institute for Basic Biology, Okazaki, Aichi 444-8585, Japan. [2] Department of Basic Biology, School of Life Science, SOKENDAI (The Graduate University for Advanced Studies), Okazaki, Aichi 444-8585, Japan. [3] Department of Biosciences and Informatics, Keio University, Yokohama, Kanagawa 223-8522, Japan. [4] Graduate School of Science, Kyushu University, Fukuoka, Fukuoka 819-0395, Japan. [5] National Institute of Genetics, Mishima, Shizuoka 411-8540, Japan. [6] Department of Biological Information, Tokyo Institute of Technology, Meguro-ku, Tokyo 152-8550, Japan. [7] Center for Ecological Research, Kyoto University, Otsu, Shiga 520-2113, Japan. [8] Faculty of Agriculture, Ryukoku University, Otsu, Shiga 520-2194, Japan. [9] RIKEN Center for Sustainable Resource Science, Yokohama, Kanagawa 230-0045, Japan. [10] Core Research for Evolutional Science and Technology (CREST), Japan Science and Technology (JST), 4-1-8 Honcho, Kawaguchi, Saitama 332-0012, Japan. [11] RIKEN Brain Science Institute, Wako, Saitama 351-0198, Japan. [12] RIKEN Center for Life Science Technologies, Yokohama, Kanagawa 230-0045, Japan. [13] Kazusa DNA Research Institute, Kisarazu, Chiba 292-0818, Japan. [14] Department of Horticulture & Life Science, Yeungnam University, Gyeongbuk 712-749, Korea. [15] Suntory Global Innovation Center Ltd, Seika, Kyoto 619-0284, Japan. [16] RIKEN Preventive Medicine and Diagnosis Innovation Program, Wako, Saitama 351-0198, Japan. [17] Department of Computational Biology, Graduate School of Frontier Sciences, The University of Tokyo, Kashiwa, Chiba 277-0882, Japan. [18] Department of Medical Genome Sciences, Graduate School of Frontier Sciences, The University of Tokyo, Bunkyo-ku, Tokyo 108-8639, Japan. [19] Principles of Informatics Research Division, National Institute of Informatics, Chiyoda-ku, Tokyo 101-8430, Japan. * These authors contributed equally to this work. † Present addresses: National Institute for Basic Biology, Okazaki, Aichi 444-8585, Japan (M.Y.); Faculty of Agriculture, Ryukoku University, Otsu, Shiga 520-2194, Japan (E.A.); Faculty of Agriculture, Meijo University, Kasugai 486-0804, Japan (Y.M.); Graduate School of Nutritional and Environmental Sciences and Graduate School of Pharmaceutical Sciences, University of Shizuoka, Shizuoka 422-8526, Japan (S.I.). Correspondence and requests for materials should be addressed to A.H. (email: hoshino@nibb.ac.jp) or to Y.S. (email: yasu@bio.keio.ac.jp).

The genus *Ipomoea*, which includes 600–700 monophyletic species, is the largest genus in the family Convolvulaceae and is a sister group to the family Solanaceae[1,2]. These species exhibit various flower morphologies and pigmentation patterns[3], and are distributed worldwide[1]. Morning glory species, including *Ipomoea nil*, *Ipomoea purpurea*, *Ipomoea tricolor* and *Ipomoea batatas* (sweet potato), are commercially important species. Japanese morning glory (*I. nil*), locally known as Asagao, is a climbing annual herb producing blue flowers capable of self-pollination (Supplementary Fig. 1a–l). It is believed to have been introduced from China to Japan in the eighth century, and has become a traditional floricultural plant in Japan since the seventeenth century. Most of Japanese elementary students grow it, as part of their school curriculum. The genetics of *I. nil* has been extensively studied for more than 100 years, and it has been a model plant for the study of photoperiodic flowering and flower colouration. A number of spontaneous mutants of *I. nil* have been identified since the early nineteenth century. Most of their mutations were related to floricultural traits, and several variants with combinations of mutations have been developed (Supplementary Fig. 1m–aa). The unique features of *I. nil*, for example, blue flowers and vine movements[4,5], have been characterized by using the cultivars carrying such mutations.

More than 1,500 cultivars of *I. nil* are maintained by the Stock Center at Kyushu University as a part of the National BioResource Project. Our recent studies have revealed that many of these mutant lines have been the result of mutagenic activity by *Tpn1* family transposons[4,6–10]. These transposons are class II elements and members of *En/Spm* or CACTA superfamily that can transpose via a cut-and-paste mechanism. The maize *En/Spm* elements encode two transposase genes for TnpA and TnpD, mediating transposition of *En/Spm* and its derivatives[11]. TnpA and TnpD bind to the sub-terminal repetitive regions (SRRs) and terminal inverted repeats (TIRs) of *En/Spm*, respectively[11]. The copy number of the *Tpn1* family was estimated to be 500–1,000, and almost 40 copies have been characterized[4,6–10,12,13]. All of the transposons characterized thus far are non-autonomous elements, and no elements encoding intact transposase genes have been identified. The non-autonomous *Tpn1* family elements have a characteristic structure and are known to capture genic regions from the host genome[12,14]. Their internal sequences are substituted with the captured host sequences, whereas their terminal regions necessary for transposition are conserved. Some of the internal genic regions are transcribed; a *Tpn1* transposon integrated in the *DFR-B* gene for anthocyanin pigment biosynthesis generates chimeric transcripts consisting of both the *DFR-B* and the captured intragenic region[14].

*I. nil* has 15 pairs of chromosomes ($2n = 30$)[15]. However, the original classical map from 1956 contained only ten linkage groups, as a result of mapping 71 genetic loci out of 219 analysed loci to one of the ten linkage groups[16]. The genetic information of *I. nil* available to date includes the linkage map[16], 62,300 expressed sequence tags (ESTs) deposited to the DDBJ/EMBL/NCBI databases, simple sequence repeat markers[17] and a recent large-scale transcriptome assembly[18]. The availability of a reference genome sequence would give researchers a standard with which to compare their mutant lines and would fast track genomic analysis of mutations. The genome of a closely related species of a wild sweet potato, *Ipomoea trifida*, was recently sequenced and published[19], in which they reported genome sequences of two *I. trifida* lines analysed using Illumina HiSeq platform, with average scaffold lengths of 6.6 kb (N50 = 43 kb) and 3.9 kb (N50 = 36 kb), respectively. However, the assembled scaffolds did not have chromosomal level information, and were highly fragmented.

In the present study, we report a pseudo-chromosomal level whole genome assembly of a wild-type *I. nil* line, with an estimated genome size of 750 Mb, sequenced using PacBio's Single Molecule, Real-Time Technology (SMRT) and Illumina sequencers. We have also identified two copies of *Tpn1* family transposons encoding putative TnpA and TnpD transposases, 339 other non-autonomous *Tpn1* transposon copies, as well as the most likely candidate for the dwarf gene, *CONTRACTED*, mapped on the classical genetic map.

## Results

**DNA sequencing and genome assembly.** One individual plant of the wild-type line, Tokyo Kokei Standard (TKS), was used for genome sequencing. Its genome size was estimated to be approximately 750 Mb using flow cytometry (Supplementary Fig. 2). PacBio sequencing yielded 5.74 million reads (39.4 GB, 52.6 × coverage and N50 of 10.3 kb), with the longest and the average read lengths being 48.1 and 6.8 kb respectively, whereas, sequencing using the Illumina HiSeq (Supplementary Table 1) included two short and six long insert libraries. With an initial read length of 150 bp, the short reads covered approximately 906 × of the genome. The work-flow for the PacBio data assembly consisted of seven steps (Supplementary Fig. 3). Initial *de novo* assembly of the PacBio reads resulted in 736.4 Mb of genome assembly, with a contig N50 of 1.83 Mb. To remove left-over residual errors originating from PacBio sequences, the short reads from Illumina were aligned against the assembled genome to identify homozygous variants. The homozygous variants amounted to 1,532 SNPs, 20,479 deletions and 6,549 insertions showing that the assembly had 99.99% base accuracy. The insertion-deletion (in-del) errors had outnumbered the substitution errors, similar to the results observed in PacBio-based *Vigna angularis*[20] and *Oropetium thomaeum*[21] genome assemblies, and were replaced with the Illumina sequence bases. Mitochondrial and chloroplast derived sequences were identified to be 1.15 Mb from 51 contigs and were removed. The organellar genomes were sequenced using a Sanger sequencer and assembled separately (Supplementary Methods; Supplementary Figs 4 and 5). Scaffolding using Illumina longer range mate-pair libraries and subsequent gap-filling using PacBio reads increased the N50 to 3.72 Mb. The assembly statistics at each step of the work-flow are mentioned in Supplementary Table 2. An independent assembly of Illumina reads using SOAPdenovo2 assembler[22] resulted in 1.1 Gb of genome assembly. The assembly size was reduced to 768 Mb, with a scaffold N50 of 3.5 Mb and a contig N50 of 9.5 kb, when considering only contigs and scaffolds longer than 1 kb. The assembly statistics of both the PacBio and Illumina assemblies are compared in Supplementary Table 3. The PacBio version of the assembly was chosen for downstream analysis owing to PacBio's longer read lengths vastly increasing the contiguity of the assembled genome.

**Mis-assembly detection and pseudo-molecule construction.** Illumina sequencing employing the RAD-seq[23] procedure, yielded 86.1 million reads for the parent samples and 562.2 million reads for the progeny samples (read length of 150 bp). Filtering the SNP markers obtained using the STACKS[24] pipeline resulted in 3,733 SNP markers from 176 samples. Fifteen linkage maps (Supplementary Fig. 6) were constructed using the SNP markers and were helpful in identifying inconsistent scaffolds which were present in more than one linkage group. To eliminate the possibility of mis-assembled chimeric scaffolds, the scaffolds were split at their junction points into two separate scaffolds using the linkage maps as a reference. In the case of mis-assemblies at the

contig level, each chimeric region was split into three parts such that the first and the last part would belong to two different chromosomes from the linkage map, whereas the middle part would still remain chimeric, albeit with a shorter length (Supplementary Fig. 7). A first splitting procedure was employed to split 52 scaffolds, after the scaffolding phase of the assembly process. After gap-filling, another splitting procedure was used to break 29 additional scaffolds. The major achievement of the assembly procedure was that, even after splitting chimeric scaffolds, the N50 values obtained for scaffolds and contigs were still 2.88 and 1.87 Mb (Table 1), respectively, which is comparable to assemblies achieved utilizing traditional Sanger sequencing data[25]. The mapping of scaffolds to linkage maps not only aided in identifying potential mis-assemblies, but also guided the generation of pseudo-chromosomes from the available scaffolds (Supplementary Data 1). The pseudo-chromosomes accounted for 91.42% of the assembly (N50 of 44.78 Mb), along

with unoriented scaffolds (around 25.53% of the assembly), and are represented in a circular display, with predicted genomic features along the 15 pseudo-chromosomes (Fig. 1a–f).

**Assembly validation.** The Core Eukaryotic Genes Mapping Approach, or CEGMA pipeline[26] and more recently, the BUSCO[27] pipeline have become commonly used protocols to validate the completeness of assembly projects by examination of coverage of highly conserved genes. The percentage of completeness for our assembly was 94.35 and 99.60% for completely and partially aligned core eukaryotic genes, respectively (Supplementary Table 4). BUSCO analysis revealed a completeness score of around 95% (Supplementary Table 5). This indicated that most of the evolutionarily conserved core gene set was present in the I. nil assembly suggesting a high quality assembly. To further validate the assembly, the newly generated I. nil ESTs, BAC-end and RNA-seq data were utilized. Comparisons against 93,691 ESTs showed that 99.11% of them were aligned, with 97.40% of the ESTs having at least 90% of their lengths covered in the alignments. Using 20,847 BAC-end read pairs, it was found that 94.92% of the reads were paired in the same scaffold with a mean insert length near the 100 kb mark (Supplementary Fig. 8), and 97.87% of the reads were paired in the same pseudo-chromosome. RNA-seq reads from six different tissues including leaf, flower, embryo, stem, root and seed coat tissues, when aligned against the assembled sequence, showed that around 94.7 and 96% of the read pairs were aligned in the embryo sample and the remaining five samples, respectively (Supplementary Table 6). The high quality of the assembly verified by CEGMA and BUSCO was corroborated by the

| Table 1 | I. nil genome assembly statistics. | | | | |
|---|---|---|---|---|---|
| Category | Total | N50 (Mb) | Longest (Mb) | Size (Mb) | Percentage of the assembly |
| Contigs* | 3,865 | 1.87 | 9.12 | 734.6 | — |
| Scaffolds | 3,416 | 2.88 | 14.4 | 734.8 | 100 |
| Anchored scaffolds | 321 | 3.14 | 14.4 | 671.7 | 91.42 |
| Genes | 42,783 | — | — | 182 | 24.77 |
| Repeats | — | — | — | 465 | 63.29 |

*The gaps in the final version of the scaffolds were split to produce the final version of contigs.

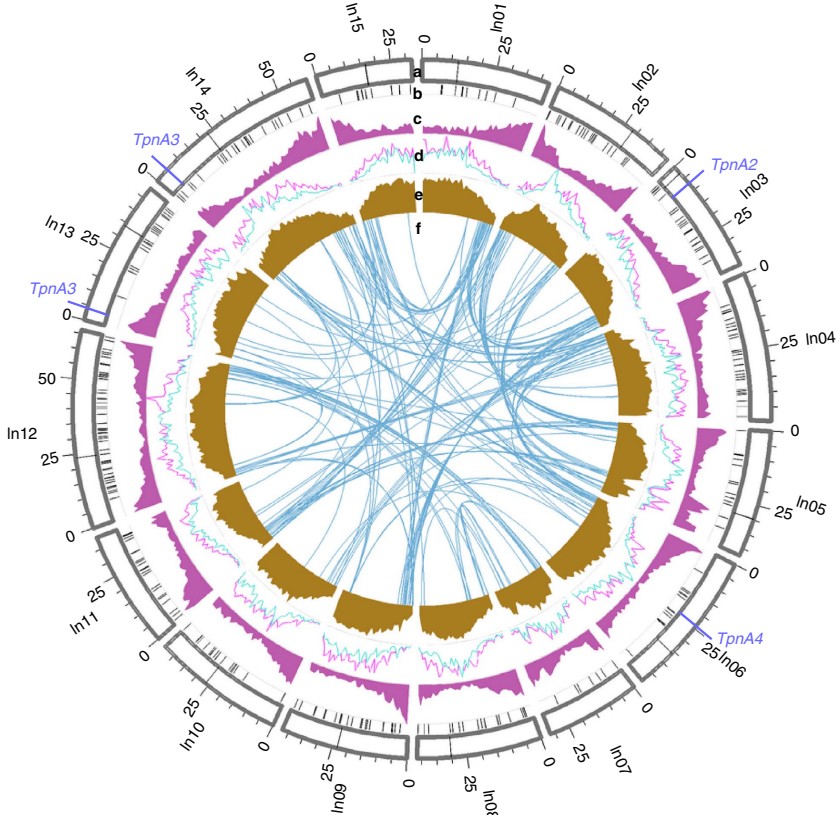

**Figure 1 | Genomic characterizations of I. nil.** (**a**) Outer circle displaying the 15 pseudo-chromosomes in 1 Mb units. *TpnA2–4* (blue dashes) and putative centromeric locations (black dashes) are also denoted in the outer circle. (**b**) Location of *Tpn1* family transposons. (**c**) Gene density per Mb. (**d**) Coverage of copia (magenta) and gypsy (turquoise) LTRs per Mb. (**e**) Repeat coverage per Mb. (**f**) Syntenic regions containing more than 10 paralogous genes.

ESTs and BAC-end sequences. Five whole BAC sequences (approximately 100 kb in length) were also completely covered in the scaffolds with minor in-dels (Supplementary Table 7). One of the BAC sequences included 12.6 kb of the *Tpn1* family transposon, *TpnA2* (see below), suggesting that repetitive elements with high copy numbers and relatively long sequences were successfully determined. The SOAPdenovo assembly was also able to cover the five BAC sequences, but with large in-dels and an increased number of mismatches, indicating that per-base resolution was better in the assembly using PacBio reads. Telomeric repeats, centromeric repeats, and rDNA arrays were identified to further analyse the contiguity of the assembly. Thirty scaffolds, with telomeric repeat units (AAACCCT) in the range of 47.1 to 4,613.9 repeating units, were identified, of which 13 were completely covered by the tandem repeats and could not be incorporated into the linkage maps (Supplementary Table 8). Pseudo-chromosomes 2, 6, 8 and 14 were found to have telomeric repeats at both the ends, while pseudo-chromosomes 3, 4, 5, 9, 10, 12, 13 and 15 had telomeric repeats at only one end. Although SOAPdenovo assembly captured 27 telomeric repeat sequences, the average size of the repeats was five times longer in the PacBio assembly. The ribosomal DNAs (rDNAs) in the order of 18S, 5.8S and 25S rDNAs are found to occur in tandem arrays typically spanning several megabase pairs in regions called nucleolar organizer regions (NORs)[21]. Three scaffolds were found to contain 3 NOR units and 34 scaffolds had 2 NOR units (Supplementary Table 9). In total, 1,212 5S rDNA sequences were clustered in 21 scaffolds that were located away from the scaffolds carrying NORs. Centromeric repeats are known to span hundreds of kilobase pairs to several megabase pairs and are difficult to be assembled owing to their repetitive complexity. The centromeric monomer sequence was identified to be 173 bp in length (Supplementary Fig. 9). Using the monomeric sequence as a base, the longest centromeric repeat stretches were identified for each chromosome and the analysis revealed that two of the identified centromeric repeat stretches were longer than 100 kb (Supplementary Table 10).

**Repeat analysis and identification of *Tpn1* transposons**. Analysis using RepeatModeler (Supplementary Table 11) showed that LTRs (long terminal repeats) comprised the largest portion of predicted repeats. The unclassified elements were mined for copia and gypsy repeats using RepBase. Copia and gypsy elements (Supplementary Table 12) comprised 12.92 and 14.46% of the assembled genome (Fig. 1d). DNA class repeat elements represented 5.60% of the genome. Altogether, 63.29% of the genome was predicted to be repetitive (Fig. 1e). However, RepeatModeler was not able to predict *Tpn1* family transposons (Supplementary Fig. 10). Hence, an in-house pipeline based on the presence of 5′ and 3′ TIRs as well as target site duplications (TSDs) was used to identify the *Tpn1* transposons. In total, 339 *Tpn1* transposons were identified with an average length of 7,081 bp (Fig. 1b; Supplementary Data 2). The smallest identified was 161 bp in length, while the longest was 40,619 bp. All the transposons had 3-bp TSDs, with the exception of one that had a 5-bp TSD (Supplementary Table 13). Fourteen of them had a mismatch in their TSDs. The TSDs tended to be AT rich (Supplementary Fig. 11), with at least one of A or T bp appearing in 95% of the TSDs. A nucleotide BLAST analysis revealed that most of the *Tpn1* transposons carried SRR sequences (Supplementary Fig. 10) in both 5′ and 3′ terminal regions (Supplementary Data 2). Because TIR and SRR sequences are *cis*-requirements for transposition, it can be suggested that the *Tpn1* transposons are capable of transposition. However, thirty-two of the identified *Tpn1* transposons contained

large rearrangements in SRR indicating that they are inactive. Twenty-nine *Tpn1* transposons were found within the 5′ UTR and introns of the predicted genes, which could disrupt the function of those genes (Supplementary Table 14). It could be expected that the autonomous *Tpn1* family transposons carry both the TnpA and TnpD transposase coding sequences such as *En/Spm* and related autonomous transposons[11]. A translated BLAST search against the 339 *Tpn1* family transposons, using TnpA and TnpD sequences from maize and snapdragon[28] as queries, revealed that two transposons, named *TpnA3* and *TpnA4*, carried *TnpD* homologues, with two copies of *TpnA3* residing in the genome (Fig. 2). No obvious *TnpA* homologues were identified in the 339 transposons. Also, no transcripts corresponding to *TnpA* and *TnpD* were found in the predicted genes or transcripts, indicating transcription of the transposase sequences was silenced in the line TKS. To identify autonomous transposons, the cDNA fragments for *TnpA* and *TnpD* homologues were isolated by a series of RT–PCRs from the line Q1072, where *Tpn1* actively transposes (Supplementary Fig. 1m). A nucleotide BLAST search, against the whole-genome sequence using the isolated cDNA sequences as queries, identified two transposons with *TnpA* and *TnpD* sequences, designated as *TpnA1* and *TpnA2* (Fig. 2). Of these, *TpnA2* is truncated in the genome, while the 5′ terminus of *TpnA1* was not completely captured in the draft genome assembly. To characterize the entire *TpnA1* sequence, a BAC clone from TKS carrying *TpnA1* was isolated and sequenced. *TpnA1* is the putative autonomous element, because it carries apparently functional TIR and SRR sequences, in addition to the coding sequences of TnpA and TnpD. No transposons carrying TnpA coding sequences alone were found. In total, the genome contained two TnpA and five TnpD putative coding sequence copies (Fig. 2). The deduced amino acid sequences of the transposases were highly conserved in the genome and shared conserved domains with known transposases of *En/Spm* and snapdragon *Tam1* (ref. 28; Supplementary Fig. 12).

**Gene prediction and functional annotation**. RNAseq data from leaf, flower, embryo, stem, root and seed coat samples were used to assist in the process of gene prediction. A total of 42,783 gene models were predicted along with 45,365 transcripts, with tomato as the reference species, using Augustus[29]. Of the transcripts, 44,916 contained a complete ORF with a start and a stop codon (Supplementary Table 15) and 95.54% of the gene models could be assigned inside the 15 pseudo-chromosomes (Fig. 1c). Single exon genes accounted for 17.52% of the total. Two thirds of the transcripts were found to have less than or equal to 5 exons. A total of 61.99% of the gene models were annotated using the UniProt-Swiss-Prot database and in the remaining gene models, 16.93% were annotated using the UniProt-Trembl database. In addition, 61.92% of the gene models were assigned Pfam domain annotations. In total, the combined annotation procedure was able to assign annotations for 79.12% of the gene models (Supplementary Data 3).

**Analysing the dwarf gene of *CONTRACTED***. The recessive *contracted* (*ct*) mutants show dwarfism with dark-green, thick, and wrinkled leaves and cotyledons[30] (Fig. 3a). Their flowers and seeds are relatively small (Supplementary Fig. 1n–p). Mutants carrying mutable *ct* alleles showing somatic reversion to the wild type were reported in the 1930s; however, existing *ct* mutants showed no mutable phenotypes[31]. The *ct* locus was mapped on LG5 of the classical linkage map and was located 1.2 cM from the *A3* locus for the anthocyanin biosynthesis gene[6,16]. It can be presumed that the *CONTRACTED* (*CT*) gene is a brassinosteroid

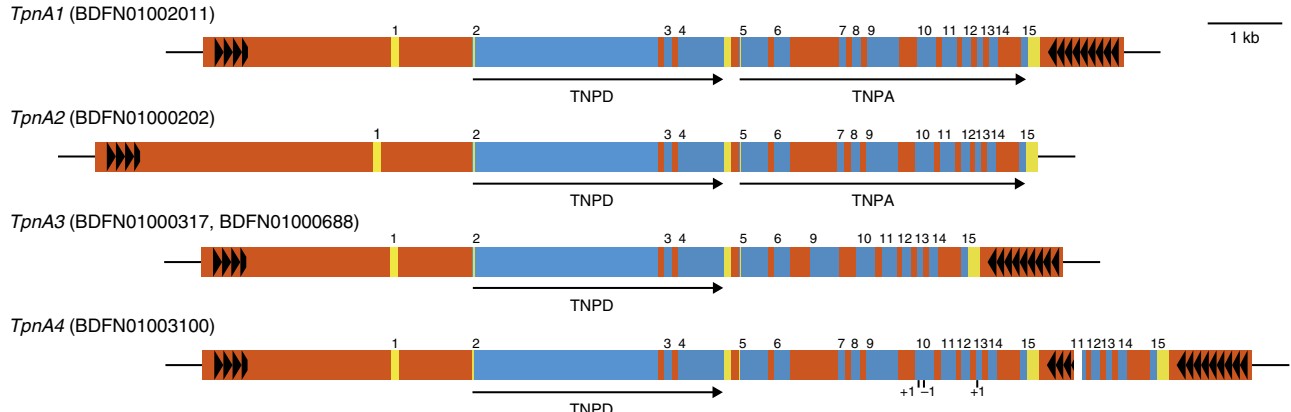

**Figure 2 | The *Tpn1* family transposons encoding transposases.** The orange, yellow and blue boxes indicate transposons, untranslated regions, and coding sequences respectively. The numerals above the blue boxes show exon numbers, and the arrows show the orientations of the transposase genes. The filled triangles are the 122-bp and 104-bp tandem repeats in the 5′ and 3′ sub-terminal regions respectively. *TpnA3* lacks exons 7 and 8, and *TpnA4* has a gap represented by a white box, as well as three frame shift mutation indicated by the vertical bar with −1 and +1 in the exon 10 and 13.

(BR) biosynthesis gene, because BR application rescued the deficiency of *ct* and *kobito* (*kbt*) double mutants[32]. The *kbt* mutation was shown to be allelic to the *star* (*s*) mutation by crossing the original *kbt* strain (Q837) and an *s* strain (Q721) (Supplementary Fig. 1r,s). The *STAR* gene is a homologue of *Arabidopsis DET2*, encoding a BR biosynthesis enzyme (Supplementary Fig. 13). To isolate the *CT* gene, BR-related genes were screened from the scaffold, BDFN01000805, carrying the *A3* gene. A homologue of *Arabidopsis ROT3* (also known as *CYP90C1*) encoding the P450 catalyzing C-23 hydroxylation of the BR precursor[33,34] was found 129 kb away from the *A3* gene in the scaffold, BDFN01000805, and its assigned gene ID was INIL05g09538 (Fig. 3b; Supplementary Fig. 14). All 19 *ct* mutants tested carried *Tpn1* family transposon insertions in the first exon of the gene, while no such insertions were found in any of the 24 lines without dwarfism (Fig. 3b; Supplementary Table 16). There are three mutant alleles of the gene named *ct-1*, *ct-2* and *ct-w*. The *ct-1* and *ct-w* alleles carried the *Tpn1* family transposons, *Tpn14* and *Tpn15*, respectively. The *ct-2* allele was derived from *ct-1*, and the 5′ half of *Tpn14* along with the 36 bp sequences immediately upstream of *Tpn14* were deleted (Fig. 3b). The transposon insertion strongly suppressed accumulation of intact mRNA of the gene in hypocotyls (Fig. 3c; Supplementary Fig. 15), whereas mRNA accumulation levels of other genes for BR biosynthesis enzyme were normal in the mutants (Supplementary Fig. 16).

**Genome evolution.** Protein sequences from rice[35] (monocotyledon outgroup), grape[36], kiwifruit[37] (from the Asterid clade), along with Solanales order members tomato[38], potato[39] and capsicum[40] were used for gene family clustering using the OrthoMCL pipeline[41] to infer phylogenetic relationships. A total of 1,353 single copy orthologs corresponding to the seven species were extracted from the clusters and were filtered to 214 single copy orthologs. Phylogenetic inference using RaxML[42] reconfirmed the phylogenetic arrangement of *I. nil* (Supplementary Fig. 17). BEAST[43] estimated the divergence of *I. nil* from the other Solanales members to be around 75.25 Myr ago, which was very close to the estimation from the TTOL[44] database (Fig. 4a). Also, *I. nil* was estimated to have separated from kiwifruit approximately 105.8 Myr ago. Divergence time estimates obtained for the other species also corresponded well with the estimations from TTOL database.

Synteny analysis using MCScanX revealed that 2,275 syntenic gene blocks were found to contain 17,376 paralogous gene pairs in the assembled pseudo-chromosomes (Fig. 1f). The number of synonymous substitutions per synonymous site (Ks) of the gene pairs in the syntenic regions was plotted against the percentage of corresponding genes to infer and compare whole-genome duplication (WGD) events in *I. nil*. *I. nil* and tomato were found to share 47.05% of syntenic orthologs in a 1:1 ratio, whereas, the percentage of kiwifruit orthologs in a 1:1 ratio across tomato and *I. nil* were 34.89 and 36.01%, respectively. Apart from the 1:1 orthologs, both tomato and *I. nil* shared large numbers of syntenic blocks with kiwifruit, possibly because of the two recent duplication events in kiwifruit[37], which was also evident from the two Ks peaks specific to kiwifruit (Fig. 4b). A recent WGD event was estimated to have occurred in Solanaceae members, approximately 71 ± 19.4 Myr ago[38,39]. A Ks peak from syntenic paralogs of tomato, corresponding to the above mentioned WGD event, was found to occur after the speciation peak between tomato and *I. nil* (Fig. 4b), suggesting that the event was specific to the Solanaceae and should have occurred reasonably close, following the divergence which was estimated to be 75.25 Myr ago (Fig. 4a). The analysis also revealed a Ks peak specific to *I. nil* indicating that a WGD event had also occurred, independently, in the Convolvulaceae family (Fig. 4b).

Gene family clustering showed that 10,549 core gene families were shared by all four species of the Solanales members (Supplementary Fig. 18). *I. nil* contained 2,242 unique gene families not shared by Solanaceae members, whereas the Solanaceae members shared 2,681 more gene families than *I. nil*. *I. nil* specific gene families had expansions of paralogs (mean value of 4.92) compared with gene families which had orthologous relationships with the other Solanales (mean value of 1.79). *I. nil* specific gene families were found to be enriched with pollination and reproductive process related gene ontology (GO) terms (Supplementary Table 17).

## Discussion

The advent of second- and third-generation sequencers have fast-tracked genome assemblies of a variety of species. The current study has utilized nearly the complete potential of recent sequencing tools and has culminated in a highly contiguous genome assembly of *I. nil*. A few of the recent genome assembly projects have used PacBio data to supplement Illumina based contig assemblies, and a mild improvement in the lengths of the assembled scaffolds have been observed (Supplementary Table 18). However, in this study, PacBio data were used as a base to construct contig assemblies, while Illumina data were used to supplement the assembly, resulting in a marked increase in the lengths of the assemblies observed (scaffold N50 length of

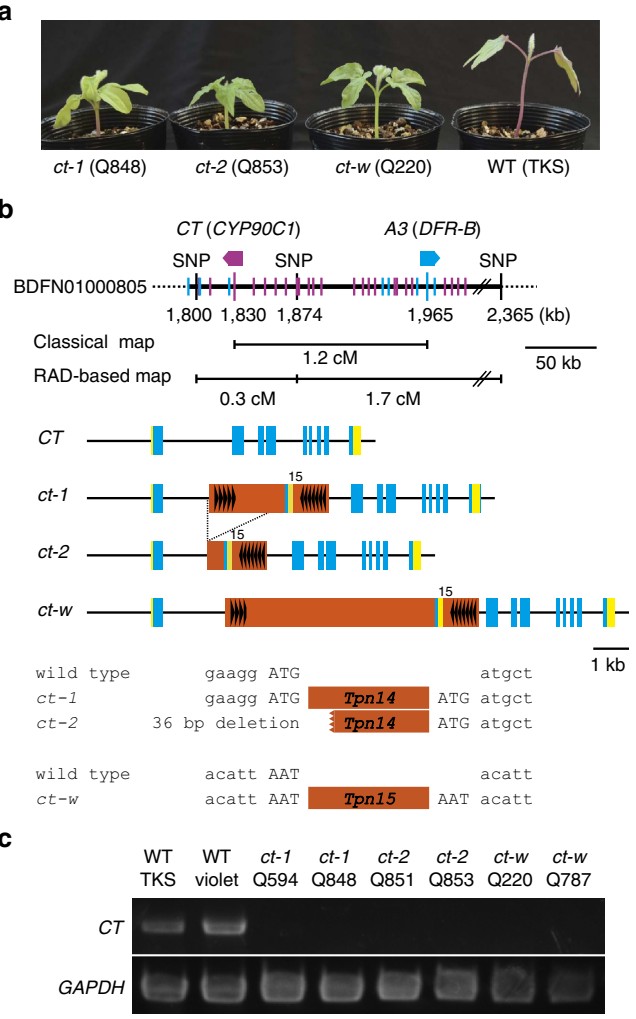

**Figure 3 | The *CT* gene for BR synthesis. (a)** Eight-day-old seedlings of the wild-type plant and the *ct* mutants. **(b)** Structure of the *CT* gene and its mutant allele. Physical map surrounding the gene (upper), genomic structure (middle) and the transposon insertion site (lower) are presented. Blue and purple bars on the physical map indicate the positions of the predicted genes with forward and reverse orientations respectively. Blue and green hexagons indicate the orientations of *CT* and *A3* respectively, and RAD indicates the nearest RAD markers. Yellow, blue, and orange boxes are untranslated regions, coding sequences, and transposons respectively. The symbols in the orange boxes are the same as in Fig. 2. **(c)** Expression of *CT* in the hypocotyls detected by reverse transcription PCR (RT–PCR).

2.88 Mb). The average contig N50 length for all published genomes is 50 kb (ref. 21), whereas *I. nil* had a contig N50 length of 1.87 Mb. The 7-kb size selected inserts of the PacBio sequence data was especially helpful in resolving *Tpn1* transposons, whose average length was approximately 7 kb, and the assembly also revealed complex repeats like telomeric repeats, rDNA clusters, and centromeric repeats. However, a better resolution of such repeats was obtained in *Oropetium thomaeum*[21] genome assembly, possibly owing to the 15-kb lower end insert size selection, explaining the importance of longer read lengths in obtaining near-perfect assemblies. The potential of PacBio sequence data in long, eukaryotic genomes has been further showcased in the draft genomes of *Gorilla gorilla*[45] (scaffold N50 of 23.1 Mb), *V. angularis*[20] (scaffold N50 of 3.0 Mb), *O. thomaeum*[21] (contig N50 of 2.4 Mb) and *Lates calcarifer*[46]

(scaffold N50 of 1.19 Mb). A rapid increase in PacBio sequencing for similar large-scale assemblies can be expected in the near future.

The draft genome has enhanced the understanding of the genetic basis of floricultural traits in *I. nil*. It was possible to catalogue *Tpn1* family transposons along with the putative autonomous element, *TpnA1* (Fig. 2; Supplementary Data 2). The *Tpn1* transposons were distributed across all 15 chromosomes (Supplementary Table 19) and one copy per 126 genes (339 copies per 42,783 genes) was observed. Most of them retain apparently functional *cis* elements, TIRs and SRRs suggesting that they are capable of transposition. In addition, *TpnA1*, *TpnA2*, *TpnA3* and *TpnA4* also encode putative transposases (Fig. 2). These features should be the basis for *Tpn1* transposons to act as the major mutagen in the mutant cultivars of *I. nil*. The *ct* mutation is traditionally called as 'uzu', and the key mutation of the barley green revolution was also named after *I. nil*'s *uzu* (contracted) because of their common semi-dwarf phenotypes[47]. It was also possible to identify the strong candidate for the *CT* gene by using the combination of the draft genome and classical linkage map, demonstrating the capability of the assembled draft genome. It can be expected that the draft genome will maximize future use of the abundant mutants and genetic knowledge of *I. nil*. Comparative analysis revealed that each of *I. nil*, tomato and kiwifruit had independent WGD events in their genomes, even though they all belonged to Asterids. One of the major reasons for the fruit-specific gene neo-functionalization in tomato is reported to be because of a large number of genes triplicated from the recent WGD event[38]. It could be assumed that the lineage specific WGDs, observed in *I. nil*, tomato and kiwifruit, could have had a major role in shaping the diverse evolution of these plant species. Being the only pseudo-chromosomal level genome assembly in Convolvulaceae, the genome sequence, linkage map and DNA clones developed in this study will facilitate not only future studies on *I. nil* and its related species, but also aid comparative genomic studies in Solanales.

## Methods

**Plant materials and sequencing.** An individual of *I. nil* Tokyo Kokei Standard (TKS) line was propagated clonally and genomic DNA isolated from the flower petals of young buds was used for whole genome sequencing. A 20 kb library (BluePippin size selection at 7 kb) for P5-C3 chemistry was constructed. Ninety SMRT cells were first sequenced on PacBio RS II system. Furthermore, sequencing libraries were prepared using the Illumina TruSeq DNA Sample Prep kit and Nextera Mate Pair Sample Prep kit. Two paired-end and six mate-pair libraries were constructed and sequenced on the Illumina HiSeq2500, with a read length of 150 bp. To validate the accuracy of the reference assembly, end sequencing of a JMHiBa BAC library was carried out using the ABI 3730xl DNA Analyzer. The TKS line was also used for construction of cDNA and BAC libraries for EST sequences. All primers used are listed in Supplementary Table 20. The genome size was estimated using a flow cytometer. For transcriptome analysis, tissues from flowers, stems, leaves, and seed coat (maternal tissue) of the individual; embryos and roots of its self-pollinated progeny were used, and the mRNA-Seq libraries were constructed using the Illumina TruSeq mRNASeq Sample Preparation Kit (version 2) from 600 ng of total RNA, collected from each of the indicated tissues, according to the manufacturer's instructions. Sequencing was conducted as paired end reads of 101 bp on Illumina HiSeq2000. Details of the sampling for the transcriptome analysis are shown in Supplementary Table 21. An F2 hybrid population of *I. nil* lines TKS × Africa (Q63) was used to construct a RAD-tag based linkage map. Two double-digested RAD libraries[17] were prepared, as described before[48] with slight modifications of the restriction enzymes and adapters. The restriction enzyme pairs were *NdeI/BglII* and *MseI/BglII* (New England Biolabs), and the adapter sequences of TruSeq MseI_NdeI and BglII adaptors are listed in Supplementary Table 20. The prepared libraries were sequenced on an Illumina HiSeq2500 platform as 151-bp single-end reads. Forty-three *I. nil* lines (Supplementary Table 16) were also used to characterize the *CT* gene. The *a3-flecked* mutant, Q1072, was used to isolate *TnpA* and *TnpD* mRNA, and an authentic *s* mutant line, Q721, was used for genetic complementation test for the *kbt* mutant, Q837.

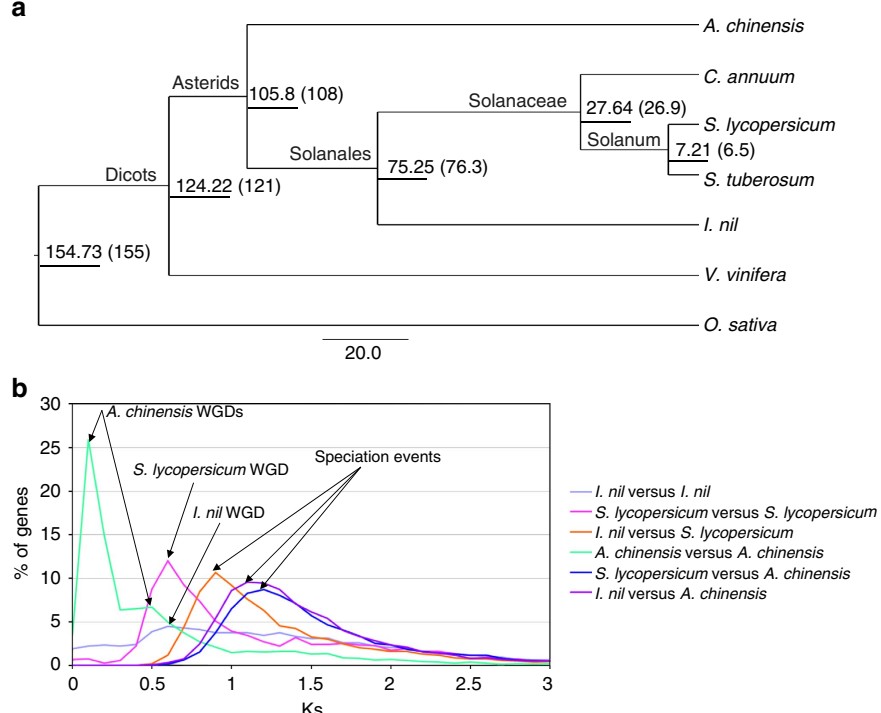

**Figure 4 | Genome evolution.** (**a**) Divergence time estimation using BEAST. The scale bar 20.0 corresponds to Myr ago. The node labels indicate estimated divergence times in Myr ago, with estimations from TTOL in parentheses, and the branch labels indicate the clades within the branch. (**b**) Distribution of Ks values against the corresponding percentage of syntenic genes, comparing *I. nil* and tomato against kiwifruit. The colours violet, magenta, orange, turquoise, blue, and purple represent the Ks values of *I. nil* versus *I. nil*, *S. lycopersicum* versus *S. lycopersicum*, *I. nil* versus *S. lycopersicum*, *A. chinensis* versus *A. chinensis*, *S. lycopersicum* versus *A. chinensis*, and *I. nil* versus *A. chinensis* respectively. Speciation events among the three species and lineage specific WGDs are highlighted with arrows.

**Genome assembly.** Before assembling the Illumina short read data set, adapters were trimmed using Cutadapt v1.2.1 (ref. 49). Using k-mer frequencies of the short insert libraries, SOAPdenovo2's error correction module (v2.01) was used to correct errors with the following parameters: '-l 80 -Q 33 -j 1 -o 3 -r 50'. The processed reads were assembled, scaffolded and gap-filled using SOAPdenovo2 assembler v2.04 (ref. 22) with a k-mer value of 115. The work-flow (Supplementary Fig. 3) of the assembly of longer PacBio reads began with contig assembly using HGAP3 pipeline[50] from SMRTanalysis v2.3.0. For HGAP3, the following parameters were used: PreAssembler Filter v1 (minimum sub-read length = 500, minimum polymerase read length = 100, and minimum polymerase read quality = 0.80); PreAssembler v2 (minimum seed read length = 6,000, number of seed read chunks = 6, alignment candidates per chunk = 10, total alignment candidates = 24, minimum coverage for correction = 6, and blasr options = 'minReadLength = 200, maxScore = 1,000, maxLCPLength = 16, and noSplitSubReads'); AssembleUnitig v1 (genome size = 750 Mb, target coverage = 30, overlap error rate = 0.06, minimum overlap length = 40, and overlapper k-mer = 14); Mapping (Maximum number of hits per read = 10, maximum divergence % = 30, minimum anchor size = 12, and pbalign options = 'seed = 1, minAccuracy = 0.75, minLength = 50, useQuality, and placeRepeatsRandomly'). The polymerase N50 and the sub-read N50 at the assembly phase was recorded as 12.3 and 10.5 kb, respectively. The initial assembly was followed by two rounds of polishing by Quiver. To correct PacBio residual errors, the Illumina reads were aligned against the contigs using BWA v0.7.12 (ref. 51). After sorting the alignments and marking duplicates using Picard tools v2.1.1 (http://picard.sourceforge.net/), Genome Analysis ToolKit v3.5 (ref. 52) was used to perform local realignment around in-dels and to call variants using the module, HaplotypeCaller. Variant filtering was performed using the expression: 'DP < 20.0||QD < 2.0||FS > 60.0||MQ < 40.0'. The homozygous in-dels were treated as errors, and were replaced with Illumina read bases in the assembled contigs using FastaAlternateReferenceMaker. MUMmer v3.23 (ref. 53) was used to identify and remove contigs, if more than 50% of their sequence was either mitochondrial or chloroplast sequence. Smaller contigs, which had greater than 98% sequence coverage in other contigs with at least 98% sequence identity, were also removed from the assembly. The contigs were then scaffolded with the help of 15 and 20 kb Illumina mate pair read libraries, with the options 'no_score and e = 10' using BESST scaffolder[54]. A first round of splitting chimeric scaffolds was performed before gap-filling. PacBio reads were utilized to gap-fill the scaffolds using PBJelly[55] with the blasr options 'minMatch = 8, minPctIdentity = 70, bestn = 1, nCandidates = 20, maxScore = 500, and noSplitSubreads'. If the flanking

sequences, at the gap junctions, had an overlap of more than 1 kb, those gaps were filled by joining the flanking sequences manually.

**Linkage map construction and pseudo-chromosome assignment.** The RAD-seq technique[23] was employed to sequence 2 parent samples (TKS and Africa lines) and 207 progeny samples. The Illumina short reads from the parent samples and progeny samples were aligned against the assembly using BWA v0.7.12 (ref. 51). The reads which were not tagged as uniquely mapped, and those which did not have the requisite restriction enzyme cut site were filtered out. STACKS v1.37 (ref. 24) was used to identify SNP and the following two criteria were used to filter markers: (a) Each marker should be present in at least 80% of the samples, and (b) Each sample should have at least 80% of the markers. Also, 150 bp flanking regions from either side of each SNP location was extracted from the assembly and was aligned against each other using BLAST to check for repetitive regions. Any region with an alignment length of longer than 150 bases were filtered out. Onemap[56] was used to create linkage maps with an LOD score of 30. TMAP[57] was used to re-order the linkage map, along with manual inspection. The original classical map contained 10 linkage groups (LGs), although *I. nil* has 15 chromosomes[15]. The marker genes from seven of the 10 LGs of the classical map[16] were mapped in the current RAD-based linkage maps, and the LGs were named 1 to 6 and 10 correspondingly (Supplementary Table 22). Because two LGs in our RAD-marker based map corresponded to LG3 in the classical map, they were accordingly assigned as LG3 and LG11 with the corresponding marker genes being *DUSKY* and *SPECKLED* respectively. This coincided with the fact that the *DUSKY* and *SPECKLED* genes were mapped on the different linkage groups in the older linkage analysis[30]. LGs 7 to 9 and 12 to 15 were numbered randomly.

**Mis-assembly elimination and assembly validation.** Before anchoring scaffolds to pseudo-chromosomes, chimeric assemblies were first resolved using linkage maps and BAC-end sequences. Contigs were first aligned against the scaffolds using the NUCmer module within MUMmer v3.23 (ref. 53) to identify the contig locations in the scaffolds. If a scaffold contained a stretch of linkage markers pointing to two different linkage groups with a scaffold junction (N) in between, it was considered a chimera and was split into two at the junction. If the mis-assembly occurred at the contig level, the bac-end alignments were used as a key in splitting chimeric contigs (Supplementary Methods). Based on the order of the linkage maps, the scaffolds were merged using Ns as gaps to form pseudo-

chromosomes. The orientations of the scaffolds were determined using the marker order, and the orientations of scaffolds with inadequate markers were ignored but included as part of the pseudo-chromosomes. The circular view of the genome was generated using Circos[58]. CEGMA v2.5 (ref. 26) and BUSCO[27], two commonly used genome assembly validation pipelines, were used to validate the completeness of genes in the assembly. BLAT was used to align ESTs and BAC-end paired reads against the assembly. In-house scripts were written, which calculated paired BLAT scores from both the BAC-end read pairs and picked up the best paired hits based on the combined score. RNA-seq reads were trimmed using Trimmomatic v0.33 (ref. 59) and TopHat v2.1.0 (ref. 60) was used to align the RNA-seq reads with default parameters. Tandem repeats finder v4.07b (ref. 61) was used to identify tandem repeats using the parameters '1 1 2 80 5 200 2000 -d -h'. Inspection of short tandem repeats at the ends of the contigs revealed the monomer 'AAACCCT' to be the telomeric repeat. Manual inspection of the tandem repeats also revealed the centromeric repeat monomer to be of approximately 173 bp in length (Supplementary Fig. 9). A tetramer centromeric repeat sequence was used to search against the whole output of tandem repeats finder using BLAST. The BLAST alignment results were screened for monomer sequences closer to 173 bp length to identify centromeric repeat candidates. Tandem centromeric repeat stretches (>3 kb) were merged, when they were within a distance of 50 kb and the longest stretch for every chromosome was identified to approximate the putative position of the centromeres. Infernal v1.1.1 (ref. 62) was used to identify rDNA clusters by searching against Rfam v12.0.

**Repeat analysis and gene prediction.** De novo repeat identification was done using RepeatModeler v1.0.7 which combines RECON and RepeatScout[63] programs, followed by RepeatMasker v4.0.2 to achieve the final results. Tpn1 family transposons were detected using the following approach: The TIRs of the Tpn1 transposons (28 bp in length) were searched using BLAST; the aligned TIR co-ordinates were sorted by their locations; if two nearby TIRs contained the same TSDs (3–5 bp), they were nominated as Tpn1 family elements. The sub-terminal repeats were also identified using BLAST to determine the orientation of the Tpn1 elements. A translated BLAST search against the identified transposons using TnpA and TnpD sequences from maize and snapdragon as queries revealed non-autonomous TpnA3 and TpnA4. To isolate autonomous Tpn1 transposons, the cDNA fragments of TnpA and TnpD homologue were isolated from Q1072 (Supplementary Methods). Using the isolated cDNA sequences as query, TpnA1 and TpnA2 were identified by screening against the assembled scaffolds using BLAST. As the 5' terminal of TpnA1 was not assembled completely in the genome sequence, a BAC clone from TKS carrying the entire TpnA1 sequence was isolated and characterized (Supplementary Methods). Repeats obtained by both the above mentioned approaches were masked for gene prediction. The genes harbouring Tpn1 transposon insertions were identified using the gene and the transposon co-ordinates and were annotated using the web version of BLASTX. Gene models were predicted using Augustus v3.2.2 (ref. 29) with tomato as the reference species, using hints from RNA-seq alignments, and also allowing prediction of untranslated regions (UTRs). Because of the scarcity of complete CDs of I. nil in public databases, independently, Augustus was also used to predict gene models, after training using CEGMA predicted genes, and the procedure resulted in more than 55,000 gene models. The 189 complete CDs sequences already available in NCBI were downloaded and compared against the predicted gene models using BLAT. Tomato based gene models showed that 116 out of 189 CDs were perfectly complete, whereas CEGMA trained gene models showed that only 61 out of 189 CDs were complete and hence, the tomato based gene predictions were used for further analysis. The gene models were translated to proteins and were aligned against proteins from UniProt-Swiss-Prot and UniProt-TrEMBL databases using NCBI BLAST + v2.2.29 (ref. 64). Using an e-value cut-off of e-5 for annotation, alignments from the Swiss-Prot database were given preference ahead of the TrEMBL database. In other words, TrEMBL annotations were assigned for only those entries without a Swiss-Prot annotation. To extract protein domain annotations, InterProScan v5.19-58.0 (ref. 65) was used to assign Pfam domains to the gene models. Gene Ontology (GO) terms were extracted from the Pfam annotations as well as UniProt annotations.

**Characterization of the CT gene.** The genomic fragments of the CT gene were amplified by PCR. The primers used are listed in Supplementary Table 20, and KOD FX Neo polymerase (TOYOBO) was used following the manufacturer's instructions. PCR fragments were purified using QIAquick PCR purification kit (QIAGEN) and were directly sequenced with the ABI Prism 3100 Genetic Analyzer and BigDye version 3.1 chemistry (Applied Biosystems). The entire genomic CT gene with the Tpn1 transposons in the six lines including Q220 (ct-1), Q787 (ct-1), Q851 (ct-2), Q853 (ct-3), Q594 (ct-w) and Q848 (ct-w) were analysed. The transposon insertion sites in the other 13 mutants listed in Supplementary Table 16 were also analysed. First strand cDNA was synthesized using ReverTra Ace qPCR RT Master Mix with gDNA Remover (TOYOBO), and cDNA fragments were amplified with KOD FX Neo using InCYP90-Fw1 and InCYP90-Rv1 primers (Supplementary Table 20). Normal expression of another CT candidate gene of INIL05g28523 encoding CYP85A2 (Supplementary Fig. 13) located more than 5.2 cM far from A3 was confirmed by RT–PCR and sequence analyses (Supplementary Fig. 16).

**Comparative analysis.** Protein sequences were downloaded from tomato, potato, capsicum, grape and rice. OrthoMCL v2.0.9 (ref. 41) was used to construct orthologous gene families, with an inflation parameter of 1.5. Before OrthoMCL, an all-vs-all BLAST was performed to find similar matches from different species, and the BLAST results were filtered with an e-value cut-off of e-5, a minimum alignment length of 50 bp, and a percentage match cut-off of 50. AgriGO[66] was used for finding GO enrichments in I. nil specific gene families. MAFFT v7.221 (ref. 67) was used for multiple sequence alignments of the resultant single copy orthologs, and trimAl v1.4 (ref. 68) was used to remove poorly aligned regions and to back-translate protein alignments to CDs alignments. The alignments were filtered using the criteria that coding sequences from each of the species should have covered at least 95% of the multiple sequence alignments, thereby, reducing the gaps to less than 5% of the alignments. RAxML v8.2.4 (ref. 42) was used to build Maximum Likelihood phylogenetic trees using the GTRGAMMA model, with rice as an out-group. BEAST v2.3.1 (ref. 43) was used to estimate the divergence times using the Jules Cantor substitution model, with a log normal relaxed clock and Yule model. The chain length of MCMC analysis was 10,000,000. TimeTree[44] is a public database containing divergence time estimates from various publications along with their own estimation. These estimates, ignoring the outliers, were used for selecting the range of lower and upper uniform calibration priors. The lower and upper calibration values were chosen as 1.9–12.8, 15.6–41, 58.6–95.1, 93.3–128.3, 101.2–156.3 and 110–216 for the most common ancestor of the seven species belonging to Solanum, Solanaceae, Solanales, asterids, dicotyledons and all plants, respectively. FigTree (http://tree.bio.ed.ac.uk/software/figtree) was used to view the phylogenetic trees. Synteny analysis of the 15 pseudo-chromosomes against the chromosomes of other species was performed using the MCScanX toolkit[69] utilizing the following options: '-m 15 -e 1e-10 -k 50'. PAML's[70] yn00 module was used to calculate the Ks values of the orthologous and paralogous gene pairs in the sytenic regions using Nei-Gojobori method. The assembled genome was compared against the genome of I. trifida (Supplementary Fig. 19).

**Data availability.** All sequencing data used in this work are available from the DNA DataBank of Japan (DDBJ) Sequence Read Archive (DRA) under the accession numbers DRA001121, DRA002710, and DRA004158 for PacBio and Illumina sequencing, DRA002647 for RNA-seq, and DRA002758 for RAD-seq. The genomic assembly sequences are available from accession numbers BDFN01000001-BDFN01003416 (scaffolds), and two organelle DNA sequences are available from accession numbers AP017303-AP017304. The EST and BAC-end sequences are available from accession numbers HY917605-HY949060 and GA933005-GA974698, respectively. Accession numbers for the CONTRACTED gene, its mutant alleles, and Tpn1 family elements are LC101804-LC101815. All the above data has been released for public access, as of August 31, 2016, and the accessibility has been verified by the authors.

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

## Acknowledgements

This work was supported by MEXT KAKENHI Grant Number 221S0002, JSPS KAKENHI Grant Numbers 22770047 and 24570062, the Model Plant Research Facility, NIBB BioResource Center, the Functional Genomics Facility, NIBB Core Research Facilities, and the Joint Usage/Research program of the Center for Ecological Research, Kyoto University.

## Author contributions

A.H., Y. Sakakibara and E.N. designed the project. A.H. and E.N. prepared plants and isolated nucleic acids. V.J. and Y. Sakakibara assembled the genome sequence. A.T., H. Noguchi, Y. Minakuchi, Y. Morita and A.F. performed genome and BAC sequencing. A.H., V.J., E.N., A.T., H.K.N., T.I. and Y. Sakakibara analysed the draft genome sequence. A.J.N., M.Y., M.N.H. and H.K. constructed RAD-seq library. A.H., M.S., A.K., T.S., P.C., E.A., S. Tabata, Y.H., K.S. and K.P. prepared the EST libraries. A.H., H. Nishide, I.U., Y.T. and S.I. analysed the EST sequences. A.H., S. Tanaka, T.S. and Y. Kohara performed cDNA sequencing. A.H., E.N., K.Y. and V.J. analysed the transposons. A.H. analysed the *CONTRACTED* gene. Y. Suzuki and S.S. performed RNA-seq. A.H., V.J., Y. Koda and Y. Sakakibara analysed RNA-seq data. V.J. performed comparative analysis, gene prediction, and functional annotation. A.H., Y. Sakakibara, E.N. and V.J. wrote the manuscript.

**Additional information**

**Competing financial interests:** The authors declare no competing financial interests.

