## [Peer Review File · Nature Communications]

Reviewers' Comments:

Reviewer #1 (Remarks to the Author)

Hoshino et al. report a high-quality, chromosome scale reference genome for the Japanese morning glory (*Ipomoea nil*) using primarily Single Molecule Real time sequencing from PacBio. They also identified a Tpn1 insertion controlling dwarfism (CONTRACTED) which was first cataloged in 1956. The *I. nil* reference genome is one of the highest-quality published to date and will be useful to the comparative genomics and sweet potato (*I. batatas*) breeding communities. This manuscript demonstrates that PacBio sequencing combined with a high-quality genetic map can generate a 'gold standard' reference genome. The manuscript is well written, the genome analyses are technically correct, and the conclusions are well supported. I feel however, there are some issues that must be resolved before this paper is suitable for publication. Below are my comments.

Major comments:

1. Were the BAC-end sequences (BES) used for scaffolding? With an average insert length of 100kb and physical coverage of $\sim 3.6x$, the BES should improve scaffolding, likely more than the Illumina mate pair libraries which have a maximum insert size of 20kb. Though the improvement might be marginal, since $\sim 96\%$ of the BES mapped to the same scaffold. The statistics of the improved PacBio/Illumina assembly should be included in supplemental table 2, including number of contigs merged, gaps filled, etc.
2. The use of SOAPdenovo's gap filling module and FGAP is potentially problematic. The long read lengths of PacBio should resolve most repetitive elements below a certain length (determined by the average read length and coverage). Any remaining gaps in the genome likely correspond to long repeat sequences or regions of high heterozygosity. With a maximum read length of 150bp, the Illumina data is unsuited for filling any remaining gaps and any gaps that were filled may be error prone. How many gaps were filled using this approach and what is the nature of these sequences?
3. The RADseq based genetic map is relatively low density (1.4 markers/ Mb) and most scaffolds are likely only anchored by a few markers. What thresholds were used to identify chimeric PacBio based contigs? If the threshold was one (potentially erroneous) marker, this could result in unnecessary contig breaking. The criteria for assembling the pseudomolecules should be better defined in the methods section.
4. Page 8: "Five whole BAC sequences (approximately 100 kb in length) were also completely covered in the scaffolds with minor insertion-deletions (In-dels) (Supplementary Table S5)."

Do these represent assembly errors in the PacBio based reference? 52x coverage is on the low end for generating a polished PacBio only assembly, the possibility of residual errors needs to be addressed. Errors can be assessed by aligning the Illumina reads to the PacBio Assembly, though this might not be necessary depending on the nature of the Indels between the BACs and the genome assembly.

Minor Comments:

1. Page 6 "... resulted in 1.1 Gb of genome assembly, with a scaffold N50 of 3.5 Mb and a contig N50 of 678 kb"

This seems to contradict the results in Supplemental Table 2 which show an average contig length of 489 bp and scaffold N50 of 2.05 Mb for the Illumina assembly.

2. What proportion of scaffolds were oriented vs anchored? The relatively low marker density would suggest most scaffolds are probably not oriented. This is shown in Supplementary figure 3 but it is difficult to read.

3. Page 16 "The work-flow (Supplementary Fig. S14) of the assembly of longer PacBio reads began with contig assembly using HGAP3 pipeline."

List the parameters used for HGAP3.

Reviewer #2 (Remarks to the Author)

In the manuscript entitled "Genome sequence and analysis of the Japanese morning glory, *Ipomoea nil*," Hoshino et al., present a high quality draft genome sequence based on a hybrid PacBio and Illumina approach, which enabled them to identify an almost complete autonomous Tpn1 En/Spm transposon and characterize the putative gene underlying CONTRACTED mutants. Morning glory is an iconic ornamental plant, and part of an important family of plants, including sweet potato. Therefore, this high quality draft genome and the analysis of the Tpn1 transposons will be of general interest. However, it is unclear from the manuscript whether the draft genome sequence (15 pseudo-chromosomes) and annotation are available for researchers to leverage, and the deposited (DDBJ) raw reads, RNAseq, RADseq, scaffolds, Tpn1 and organellar sequence were not available to this reviewer to validate the claims presented. An important aspect of this work that will make it of general interest is the availability of the assembled genome and annotation for the scientific community to leverage and build additional findings upon.

The authors utilize PacBio sequence to generate the draft genome sequence, which resulted in a final assembly with excellent contig contiguity statistics with a N50 length of 3.25 Mb. This is a very typical result for a plant PacBio assembly using HGAP3 for a genome this size, but as the authors note, this a great contig N50 length compared to other published plant genomes (even Sanger based assemblies). They then utilize a very large amount (900x) of Illumina paired-end and mate-pair data to improve contiguity through scaffolding and gap filling the PacBio contigs. However, the assembly only improves incrementally with a scaffold N50 length 4.5 Mb, which is an average scaffold N50 length and much lower than some of the gold standard draft genomes. However, at this point, the genome assembly is far superior to the closely related published sweet potato genome(s). The authors finish off the assembly by leveraging RADseq to place the scaffolds on a linkage map, and break miss-assemblies. The result is a final draft genome assembly in 15 linkage groups with a great contig N50 length.

Since a highlight of the data presented is the quality of the draft genome assembly, it would be great to know what the assembly statistics are at each round of the hybrid assembly. For instance, what were the assembly statistics of the HGAP3 assembly alone? Also, what were the polymerase and subread N50 length of the PacBio data going into the assembly? Was the HGAP3 assembly polished with quiver? Table S2 should include the contig N50 length information. Since the scaffold N50 was a modest improvement over the PacBio assembly it is not clear that adding the Illumina data was the best approach, but this could only be assessed by looking at the assembly at each stage. In general it would be better practice to assess the quality of the PacBio assembly and polish contigs using the Illumina data. Then utilize the RADseq data to anchor the contigs. While 10, 15 and 20 kb Illumina mate-pair libraries are available they clearly were not effective in crossing the repeat regions and are best used to validate the PacBio assembly. A better way to report the scaffold N50 length (L50) after consolidation into linkage groups with RADseq would be as super-scaffolds, which should be about 40 Mb with gaps based on Supplementary Data S1 (Access to the genome assembly would have made this very quick to check).

The authors generate an Illumina only assembly using SOAPdenovo2, but the genome size is ~400 Mb

larger than the hybrid PacBio/Illumina assembly (736,250,312 vs. 1,106,449,450 bp). How do the two assemblies compare in terms of per base accuracy and quality? How does the hybrid assembly compare to the PacBio alone and the Illumina alone assemblies? Contiguity (L50) is typically one of the most widely used metrics for genome assembly quality, but now that more PacBio-only assemblies are available it is becoming clear that per base and local accuracy are an issue. In addition, HGAP3 is very good at assembling some types of repeats while failing at others, so does the Illumina only assembly expose these issues? How well did the rDNA clusters assemble, and how many arrays are found in the *I. nil* genome. Is the 5S cluster located on a separate linkage group? How well did the centromeres and telomeres assemble? What is the relationship between the transposable elements (repeats in general) and the centromeres in the *I. nil* genome?

While the authors did not use the Illumina data to validate the assembly these did check the assembly by mapping ESTs, BACs and RNAseq as well as the CEGMA tool. The EST, BAC and RNAseq data are consistent with the level of contiguity of the draft genome assembly (Figure S5 is not very informative and adds nothing to the supplement). However, CEGMA predicts 94% complete; how does this compare to other draft genomes? If the genome is as high quality as the authors state, and CEGMA is predicting only 94% complete, what could this mean about *I. nil* gene predictions? For reference, *Arabidopsis* (TAIR10) is 99.2% complete. To this end, the authors predict 41 k genes using only one *ab initio* gene prediction program (Augustus). While 41 k genes may not be that high, it still is higher than most plant genomes and could reflect erroneous gene prediction. Small RNAs are predicted but not compared to other plant genomes. How do these compare to sweet potato, other plant genomes, and Solanales? Is there anything new or unexpected about the *I. nil* genome? With the "first pseudo-chromosomal assembly in Convolvulaceae" is there anything special or distinct about this family? Considering the interesting stem behavior and the prominence of the flowers are there anything interesting in regard to the genes that are involved in these processes?

I. nil is closely related to the parasitic plant *Cuscuta*, which has significant variation around chloroplast content, size and presence/absence. The authors state that the organellar genomes are deposited at DDBJ under AP017303-AP017304, but they do not provide any analysis in either the main text or supplemental material. Generally PacBio sequence data enables complete assembly of the chloroplast genome and if not the entire mitochondria, it will be in several contigs (usually due to linear or multiple small circles). A statement of the completeness and an analysis of the organellar genomes would add significantly to the genome assembly and analysis.

Overall, the authors have presented a very lite analysis of the *I. nil* genome, with a cursory whole genome duplication (WGD) analysis thrown in at the end. However, the main story is about the Tpn1 transposable elements, which have made morning glory both a fantastic model system as well as ornamentally beautiful. This is where taking the PacBio approach to sequence the genome makes a lot of sense to get the repeat regions; although, the authors don't make a point of this in the text. The high contiguity draft genome assembly is very useful at identifying Tpn1-like transposons, and the authors find 322. However, they have to use a cloned version of the Tpn1 genes from another line Q1072 to find an almost complete autonomous Tpn1 element. This part of the manuscript is very hard to interpret because it is not well referenced or described. It is not clear why the Tpn1A did not come out of the original search because based on Figure 2 it seems to have the sequence that was used to identify the 322 Tpn1 non-autonomous sequences. Furthermore, they validate and find the complete sequence of Tpn1A with a sequenced BAC (some of Tpn1A is missing in the draft genome assembly); this suggests the genome was not necessary to find Tpn1A and the old fashion way would have worked first and been more effective. Also, Supplementary Data S2 (Predicted Tpn1 transposons) reports the 322 identified non-autonomous Tpn1 transposons. Using this information did the authors find any transposon "scabs" in the TKS reference genome; possibility disrupting coding regions, or islands in the pericentric regions where a history of transposition could be followed?

Finally, the authors clone the CONTRACTED (CT) gene using the genome reference and the classical map position of the mutant. Based on the map position, and the hypothesis that the gene is involved in brassinosteroid biosynthesis they suggest the CT mutation is due to several different types of Tpn1 insertions into the homologue of the Arabidopsis ROT3 gene. Once again, a genome sequence was not required to identify the CT gene but this is a great use of the draft genome sequence to do reverse genetics. It was not clear from the text or the references whether the mutants analyzed (Table S11) were known previously or whether this was a completely new analysis (sequencing of each line?) presented here. This section is important so the text needs to be clearer as to what was actually done in this study; even the methods section is brief and does not clarify what was actually done here. Moreover, what other genes are candidates in this region (can't tell from Supplementary Data 3 because there is no way to cross reference scaffolds and gene numbers; what is the ROT3 homologue I. nil gene number?) and were any other genes tested for potential Tpn1 insertions? While, it seems probable that ROT3 is the CT mutant gene, no validation is provided so it is formally possible another linked gene plays a role. If the number of Tpn1 insertions is ~1000 per genome then there is a chance of another insertion in this region that could explain the phenotype. How can you rule this out? How many non-autonomous Tpn1 transposons are located in this region in the TKS reference?

Minor suggestions:

Add genome size to the abstract and introduction.

Claims of being the "first" are not necessary if the results are significant; remove from the abstract and introduction.

Page 7. "In case of mis-assemblies at contig level," should read "In the case of mis-assemblies at the contig level."

Page 7. "...achieved utilizing traditional Sanger sequencing data." Needs a reference.

Page 9. Reporting the 3:2 copia to gypsy ratio as 3:2 is misleading since the 22% represents uncharacterized copia and gypsy elements specific to I. nil, which could be updated by annotating the unclassified.

Page 9. "...TpnA and TnpD transposase coding sequence." Needs a reference.

Differential expression is presented in Supplementary Data S4 (Scaled FPKM values of clustered DEGs); are there replicates for that data? If not then what does this data really mean?

Page 14. *Vigna angularis* and *Oropetium thomaeum* are mentioned in the discussion but they do not appear in Table S12.

Reviewer #3 (Remarks to the Author)

I have reviewed the manuscript, "Genome sequence analysis of the Japanese morning glory, *Ipomoea nil*", by Hoshino et al. This is a basic genome sequencing paper with some relevant analysis about transposable elements content and about comparative genomic analyses with other sequenced dicot species. This paper was a pleasure to read. It is well written, and I appreciate that a great deal.

The authors used PacBio sequences to generate their genome assembly. In fact, they also performed Illumina sequencing, but the resulting assembly was not quite as good as the PacBio assembly. The Illumina sequence was used for assembling scaffolds from the PacBio unitigs and for gap filling. This

procedure was explained well, and the Illumina vs PacBio comparison is very useful. If you have a budget that will allow PacBio sequencing, an impressive genome assembly can be obtained even for a medium sized genome. Otherwise, you can have a perfectly functional genome assembly for a more affordable cost.

I do have issues/concerns.

The authors used CEGMA to assess the completeness of their genome assembly. CEGMA is no longer the standard for this assessment. BUSCO is the new tool that should be used for this purpose. Even the author of CEGMA has indicated that BUSCO should be the standard tool for assessing genome completeness. In fact, it is acceptable to also report CEGMA results, but as it is relatively easy to run, the authors should use BUSCO on their genome assembly. The results from BUSCO will not show 90+% completeness. Even rice and arabidopsis only get BUSCO scores in the 80's. Even without BUSCO results, the other controls that the authors have included indicate that the assembly is very good.

I am really dissatisfied that the authors used the Augustus tomato HMM for their gene prediction. This is very bad practice. In the article that describes the SNAP gene annotation program, Ian Korf demonstrates why using an HMM from one species to predict genes in a second species is a bad idea. The utility of a SNAP HMM to predict genes in a new genome is not correlated with phylogenetic similarity between the species used to train the HMM and the target species. The utility of a SNAP HMM to predict genes in a new genome is correlated with the GC content of the genes in the two species. I am very close to nixing this submission over this issue. Gene prediction HMMs do not accurately predict genes with GC content that is wildly different than the GC content of the genes that had been used to train the HMM. Predictions will be made, but they are often not accurate. Augustus should have been trained with example genes from *Ipomoea nil*. The process is obscure but not difficult. The Augustus developers will even help with the training. I would like the authors to calculate the GC content of the CDSes from *Ipomoea nil* ESTs and/or fl-cDNAs. A couple hundred of these sequences would be sufficient. The GC content of the predicted CDSes from the tomato genome should also be calculated. These two GC content values should be very similar (within about 5%) to be able to argue that using the August tomato HMM was acceptable. I think that a difference in these GC values of more than 10% would cause me to be concerned about the gene annotation. Another number that would help to indicate that the *Ipomoea nil* genes were well annotated would be the total number of genes that had been in an OrthoMCL ortholog group with at least one other species. I don't think that this number was presented.

The tRNA results were very interesting. However, I have been taught that if a result is not interesting enough to mention in the Discussion section, then it does not need to appear in the Results section. I would like to see some analysis of this tRNA finding in the Discussion section. Are there some other analyses beyond the Infernal analysis that could be run to support the identification of so many tRNAs? Wow. This is out of control if true.

In the discussion, the authors state that Tpn1 transposons are the major mutagen in *Ipomoea nil*. That is a pretty emphatic statement that I don't think has been sufficiently supported. Are Tpn1 transposons more active than GC-biased gene conversion or 5-methylcytosine to thymine conversion? I cannot let this one slide. Something has to change in the text.

REVIEWERS' COMMENTS:

Reviewer #1 (Remarks to the Author):

The authors have adequately addressed my concerns and I feel the revised manuscript is suitable for publication.

Reviewer #2 (Remarks to the Author):

As a point of clarification, this reviewer wanted to have access to the genome to validate quality and claims. Since the genome was not available for scrutiny by the reviewers, the claims concerning the genome will have to be taken at face value.

The authors have addressed my suggestions and concerns, and the modifications and additions add value to the manuscript.

One small sticking point brought up by one of the other reviewers is the gene prediction based on tomato gene models. The authors make a good case for using the tomato models in their rebuttal. However, based on the information in the introduction, there seems to be ample information to make *I. nil* models:

Lines 103-105

"62,300 expressed sequence tags (ESTs) deposited to the DDBJ/EMBL/NCBI databases, Simple Sequence Repeat (SSR) markers and a recent large scale transcriptome assembly".

And

Lines 182-184

"Comparisons against 93,691 ESTs showed that 99.11 % of them were aligned, with 97.40 % of the ESTs having at least 90 % of their lengths covered in the alignments.

A path forward would be to make it very clear that tomato models were used by adding a statement in lines 262-263 that is similar to what is in the methods section:

Line 510

"Gene models were predicted using Augustus v3.2.229 with tomato as the reference species,..."

Also, the rebuttal explanation (or a concise version of it) would add value and clarity in the methods section.

Minor editorial suggestions

Lines 101-103

It would flow better if these two sentences were edited to make it clear that there are 15 chromosomes and 10 of them have mutants mapped to them.

I. nil has 15 pairs of chromosomes ($2n = 30$)¹⁵. A total of 219 genetic loci of *I. nil* had been analyzed by 1956. Among them, 71 loci were mapped to one of the 10 linkage groups (LGs).

Lines 108-111

Should read "average scaffold length."

The genome of a closely related species of a wild sweet potato, *I. trifida*, was recently sequenced 109

and published, in which they reported genome sequences of two *I. trifida* lines analyzed using Illumina HiSeq platform, with an average length of 6.6 kb (N50 = 43 kb) and 3.9 kb (N50 = 36 111 kb), respectively.

Lines 302-303

214 what? Restructure the sentence to make it clear as to what 214 refers to.

A total of 1,353 single copy orthologs corresponding to the seven species were extracted from the clusters and were filtered to 214.

Lines 336-337

Highly contiguous would sound better than long because a long genome assembly does not make sense.

"variety of species. The current study has utilized nearly the complete potential of recent sequencing tools and has culminated in a long, high quality genome assembly of *I. nil*."

Line 784

Figure legend 4, define the color and lines within the text of the legend.

Reviewer #3 (Remarks to the Author):

I have read both the revised version of "Genome sequence and analysis of the Japanese morning glory, *Ipomoea nil*" as well as the comments to reviewers. The current version of the manuscript reads very well. The authors have also addressed my specific comments to my satisfaction.

Dear Reviewers,

We greatly appreciate the reviewers' constructive comments, which helped us to considerably improve the quality of our manuscript. The manuscript has been revised accordingly.

In summary, the main revisions include:

1. Initial *de novo* assembly using HGAP pipeline was followed by scaffolding using only the 15k and 20k insert Illumina libraries (instead of the previous version, wherein all the eight Illumina libraries were used for scaffolding).
2. The gap-filling procedure, using shorter Illumina reads and scaffolds, was replaced with gap-filling using longer PacBio reads with PBJelly.
3. Residual error correction in the PacBio assembly using Illumina reads.
4. Removal of the gene expression analysis owing to lack of replicates.
5. Removal of non-coding RNA analysis.
6. In addition, minor revisions recommended by the reviewers have been incorporated. Some of them include BUSCO analysis, identification of telomere repeats, centromere repeats, and rDNA clusters.
7. Annotation of the organelle genomes.
8. Corresponding text in the manuscript has been edited accordingly.

[Comments of Reviewer #1:]

[Comment of Reviewer #1:]

Hoshino et al. report a high-quality, chromosome scale reference genome for the Japanese morning glory (*Ipomoea nil*) using primarily Single Molecule Real time sequencing from PacBio. They also identified a *Tpn1* insertion controlling dwarfism (CONTRACTED) which was first cataloged in 1956. The *I. nil* reference genome is one of the highest-quality published to date and will be useful to the comparative genomics and sweet potato (*I. batatas*) breeding communities. This manuscript demonstrates that PacBio sequencing combined with a high-quality genetic map can generate a 'gold standard' reference genome. The manuscript is well written, the genome analyses are technically correct, and the conclusions are well supported. I feel however, there are some issues that must be resolved before this paper is suitable for publication.

Response: We thank the reviewer for the positive and encouraging comments, and we are glad that the importance of the manuscript was aptly recognized.

[Comment of Reviewer #1:]

Below are my comments.

Major comments:

1. Were the BAC-end sequences (BES) used for scaffolding? With an average insert length of 100kb and physical coverage of ~3.6x, the BES should improve scaffolding, likely more than the Illumina mate pair libraries which have a maximum insert size of 20kb. Though the improvement might be marginal, since ~96% of the BES mapped to the same scaffold.

Response: As rightly stated by the reviewer, the physical coverage of BAC-end sequences is ~3.6X. As suggested, we tried scaffolding the contigs utilizing BAC-end sequences with a read depth cut-off of 3. However, linkage map analysis showed an increase in possible scaffold mis-assemblies, and therefore, we decided not to utilize the BAC-end sequences in scaffolding.

[Comment of Reviewer #1:]

The statistics of the improved PacBio/Illumina assembly should be included in supplemental table 2, including number of contigs merged, gaps filled, etc.

Response: We appreciate the need to separate the statistics from the main text. Thank you for the suggestion. We have included the stepwise assembly statistics in the revised manuscript (Please refer to Supplementary Table S2).

[Comment of Reviewer #1:]

2. The use of SOAPdenovo's gap filling module and FGAP is potentially problematic. The long read lengths of PacBio should resolve most repetitive elements below a certain length (determined by the average read length and coverage). Any remaining gaps in the genome likely correspond to long repeat sequences or regions of high heterozygosity. With a maximum read length of 150bp, the Illumina data is unsuited for filling any remaining gaps and any gaps that were filled may be error prone. How many gaps were filled using this approach and what is the nature of these sequences?

Response: As stated by the reviewer, we realize that gap-filling using shorter reads introduces the risk of error-prone assembly, and therefore, we have replaced this approach with gap-filling using PacBio reads with PBJelly.

[Comment of Reviewer #1:]

3. The RADseq based genetic map is relatively low density (1.4 markers/ Mb) and most scaffolds are likely only anchored by a few markers. What thresholds were used to identify chimeric PacBio

based contigs? If the threshold was one (potentially erroneous) marker, this could result in unnecessary contig breaking. The criteria for assembling the pseudomolecules should be better defined in the methods section.

Response: When there are at least 2 markers in a scaffold corresponding to two different pseudo-chromosomes from the linkage map, we had induced a breakpoint between them. As the reviewer pointed out, some may lead to unnecessary contig breaking. However, our conservative strategy will result in fewer mis-assemblies, although the tradeoff is a shortening in the read lengths. We have revised the text in the Supplementary Methods section for better clarity.

[Comment of Reviewer #1:]

4. Page 8: "Five whole BAC sequences (approximately 100 kb in length) were also completely covered in the scaffolds with minor insertion-deletions (In-dels) (Supplementary Table S5)." Do these represent assembly errors in the PacBio based reference? 52x coverage is on the low end for generating a polished PacBio only assembly, the possibility of residual errors needs to be addressed. Errors can be assessed by aligning the Illumina reads to the PacBio Assembly, though this might not be necessary depending on the nature of the Indels between the BACs and the genome assembly.

Response: In the revision, we have included a correction step using Illumina reads for residual errors. As suggested, homozygous indels and SNPs were identified by aligning the Illumina reads against the assembled genome. The homozygous indels were replaced with Illumina bases to eliminate the possibility of residual errors. The per-base accuracy was estimated to be 99.99%, considering homozygous variants. However, we still find a few small in-dels (although reduced compared to the assembly without error-correction) in the whole BAC sequences in comparison to the assembly (Supplementary Table S7). When the whole BAC sequences were aligned against SOAPdenovo assembly, we found that there was an increased number of mismatches and large in-dels, suggesting that the per-base accuracy is much better in PacBio based assembly.

[Comment of Reviewer #1:]

Minor Comments:

1. Page 6 "... resulted in 1.1 Gb of genome assembly, with a scaffold N50 of 3.5 Mb and a contig N50 of 678 kb"

This seems to contradict the results in Supplemental Table 2 which show an average contig length of 489 bp and scaffold N50 of 2.05 Mb for the Illumina assembly.

Response: One of the SOAPdenovo assembly results corresponded to the statistics obtained after removing scaffolds less than 1 kb. We apologize for the confusion. We have corrected this in the revised manuscript.

[Comment of Reviewer #1:]

2. What proportion of scaffolds were oriented vs anchored? The relatively low marker density would suggest most scaffolds are probably not oriented. This is shown in Supplementary figure 3 but it is difficult to read.

Response: We managed to increase the markers by a small extent by discarding the earlier method. In the earlier method, we directly identified SNP markers in the repeat masked genome. In the current process, SNP markers were identified in the non-masked genome, and the flanking sequences were then aligned against each other to identify and remove markers from the repetitive sites. Currently, approximately 91% of the assembly is anchored, while 25% is un-oriented.

[Comment of Reviewer #1:]

3. Page 16 "The work-flow (Supplementary Fig. S14) of the assembly of longer PacBio reads began with contig assembly using HGAP3 pipeline."

List the parameters used for HGAP3.

Response: We have listed the following parameters in the revised manuscript.

PreAssembler Filter v1 (minimum sub-read length = 500, minimum polymerase read length = 100, and minimum polymerase read quality = 0.80); PreAssembler v2 (minimum seed read length = 6000, number of seed read chunks = 6, alignment candidates per chunk = 10, total alignment candidates = 24, minimum coverage for correction = 6, and blasr options = “minReadLength = 200, maxScore = 1000, maxLCPLength = 16, and noSplitSubReads”); AssembleUnitig v1 (genome size = 750 Mb, target coverage = 30, overlap error rate = 0.06, minimum overlap length = 40, and overlapper k-mer = 14); Mapping (Maximum number of hits per read = 10, maximum divergence % = 30, minimum anchor size = 12, and palign options = “seed = 1, minAccuracy = 0.75, minLength = 50, useQuality, and placeRepeatsRandomly”). The initial assembly was followed by two rounds of polishing by Quiver.

We thank the reviewer once again for the constructive comments.

[Comments of Reviewer #2:]

[Comment of Reviewer #2:]

In the manuscript entitled "Genome sequence and analysis of the Japanese morning glory, *Ipomoea nil*," Hoshino et al., present a high quality draft genome sequence based on a hybrid PacBio and Illumina approach, which enabled them to identify an almost complete autonomous Tpn1 En/Spm transposon and characterize the putative gene underlying CONTRACTED mutants. Morning glory is an iconic ornamental plant, and part of an important family of plants, including sweet potato. Therefore, this high quality draft genome and the analysis of the Tpn1 transposons will be of general interest. However, it is unclear from the manuscript whether the draft genome sequence (15 pseudo-chromosomes) and annotation are available for researchers to leverage, and the deposited (DDBJ) raw reads, RNAseq, RADseq, scaffolds, Tpn1 and organellar sequence were not available to this reviewer to validate the claims presented. An important aspect of this work that will make it of general interest is the availability of the assembled genome and annotation for the scientific community to leverage and build additional findings upon.

Response: We thank the reviewer for summarizing the work and highlighting the important points in the study. As with other genome studies, we chose DDBJ's option of delaying the public release until publication. It also ensures that we make the correct data public. For instance, we had made major revisions to this study after the review process. The purpose of this study is of course to make the data publicly available so that a large community of researchers in the field of *I. nil* research could be benefited. We assure that the data would be public as soon as the manuscript is accepted for publication.

[Comment of Reviewer #2:]

The authors utilize PacBio sequence to generate the draft genome sequence, which resulted in a final assembly with excellent contig contiguity statistics with a N50 length of 3.25 Mb. This is a very typical result for a plant PacBio assembly using HGAP3 for a genome this size, but as the authors note, this a great contig N50 length compared to other published plant genomes (even Sanger based assemblies). They then utilize a very large amount (900x) of Illumina paired-end and mate-pair data to improve contiguity through scaffolding and gap filling the PacBio contigs. However, the assembly only improves incrementally with a scaffold N50 length 4.5 Mb, which is an average scaffold N50 length and much lower than some of the gold standard draft genomes. However, at this point, the genome assembly is far superior to the closely related published sweet potato genome(s). The authors finish off the assembly by leveraging RADseq to place the scaffolds on a linkage map, and break miss-assemblies. The result is a final draft genome assembly in 15 linkage groups with a great contig N50 length. Since a highlight of the data presented is the quality of the draft genome assembly, it would be great to know what the assembly statistics are at each round of the hybrid

assembly. For instance, what were the assembly statistics of the HGAP3 assembly alone?

Response: The contig results presented in the tables are not the original results of the HGAP *de novo* assembly process, but instead, are the contiguous sequences obtained after splitting the gaps (Ns) in the scaffolds. To avoid confusion, we have included the statistics of the step-wise assembly improvements in Supplementary Table S2. The table also bifurcates the contig and scaffold statistics at each phase. As shown, the N50 increased from 1.8 Mb in contigs to 4 Mb in scaffolds.

[Comment of Reviewer #2:]

Also, what were the polymerase and subread N50 length of the PacBio data going into the assembly? Was the HGAP3 assembly polished with quiver?

Response: Polymerase N50 is 12,374 bp and sub-read N50 is 10,564 bp. Yes, we had included 2 rounds of polishing by quiver.

[Comment of Reviewer #2:]

Table S2 should include the contig N50 length information. Since the scaffold N50 was a modest improvement over the PacBio assembly it is not clear that adding the Illumina data was the best approach, but this could only be assessed by looking at the assembly at each stage. In general it would be better practice to assess the quality of the PacBio assembly and polish contigs using the Illumina data. Then utilize the RADseq data to anchor the contigs. While 10, 15 and 20 kb Illumina mate-pair libraries are available they clearly were not effective in crossing the repeat regions and are best used to validate the PacBio assembly. A better way to report the scaffold N50 length (L50) after consolidation into linkage groups with RADseq would be as super-scaffolds, which should be about 40 Mb with gaps based on Supplementary Data S1 (Access to the genome assembly would have made this very quick to check).

Response: The initial N50 obtained after HGAP assembly was 1.8 Mb. After scaffolding and after splitting mis-assemblies, the N50 length of the final scaffolds increased considerably to 2.8 Mb in the revised post-assembly workflow. As suggested, we had used Illumina data to polish the PacBio assembly. We had also used the RADseq data to anchor the scaffolds into super-scaffolds (pseudo-chromosomes). Yes, it is closer to 40 Mb (44.78 Mb, Supplementary Table S19), as calculated by the reviewer, and we have also included the information in the manuscript.

[Comment of Reviewer #2:]

The authors generate an Illumina only assembly using SOAPdenovo2, but the genome size is ~400 Mb larger than the hybrid PacBio/Illumina assembly (736,250,312 vs. 1,106,449,450 bp). How do the two assemblies compare in terms of per base accuracy and quality? How does the hybrid assembly compare to the PacBio alone and the Illumina alone assemblies? Contiguity (L50) is typically one of the most widely used metrics for genome assembly quality, but now that more PacBio-only assemblies are available it is becoming clear that per base and local accuracy are an issue. In addition, HGAP3 is very good at assembling some types of repeats while failing at others, so does the Illumina only assembly expose these issues?

Response: The transposons of *I. nil* are generally in the range of kilo bases, and are abundant. This fact should be reflected in the Illumina alone assembly with largely fragmented contigs (Contig N50 < 10 Kb and the number of contigs = 2,262,957). In terms of per base accuracy identified using Illumina data and homozygous variants, the PacBio assembly had 99.99 % accuracy. Owing to limited genomic information for *I. nil*, it becomes difficult to accurately estimate per base accuracies for SOAPdenovo assembly. With the 5 BAC sequences analyzed by Sanger sequencer, we found that the PacBio sequences had completely covered them with minor indels, whereas with SOAPdenovo assembly, the assembly had an increased number of mismatches and large in-dels. In terms of repeats, when we compared the completion of telomere repeats, the PacBio assembly was able to capture 30 telomeric repeat sequences corresponding to 15 chromosomes. Although SOAPdenovo assembly captured 27 telomeric sequences, the average size of the repeats was 5 times longer in the PacBio assembly. These features indicate that PacBio is much better at

assembling difficult parts of the genome.

[Comment of Reviewer #2:]

How well did the rDNA clusters assemble, and how many arrays are found in the *I. nil* genome. Is the 5S cluster located on a separate linkage group? How well did the centromeres and telomeres assemble? What is the relationship between the transposable elements (repeats in general) and the centromeres in the *I. nil* genome?

Response: Thirty telomere repeats corresponding to 15 chromosomes were identified in the assembly (Supplementary Table S8), although only 16 of them could be allocated within pseudochromosomes. The assembly could identify rDNA arrays and centromere repeats successfully (Supplementary Tables S9 and S10). In all, three scaffolds were found to contain 3 NOR units and 34 scaffolds had 2 NOR units, and 1,212 5S rDNA sequences were clustered in 21 scaffolds. Both the 5S and NORs were identified in separate linkage groups. The longest centromeric repeat stretches were identified for each chromosome and the analysis revealed that two of the identified centromeric repeat stretches were longer than 100 kb (Supplementary Table S10). However, the size selection of 7-kb inserts was probably not sufficient. A better resolution of such repeats was obtained in *Oropetium thomaeum* genome assembly, possibly owing to the 15-kb lower end insert size selection, explaining the importance of longer read lengths in obtaining near-perfect assemblies. We were able to only putatively approximate the centromere locations and hence, we did not attempt to analyze the relationships of centromere repeats and TEs in detail. However, from Fig. 1d-e, it could be understood that the putative centromere locations are generally repeat rich and gene poor regions.

[Comment of Reviewer #2:]

While the authors did not use the Illumina data to validate the assembly these did check the assembly by mapping ESTs, BACs and RNAseq as well as the CEGMA tool. The EST, BAC and RNAseq data are consistent with the level of contiguity of the draft genome assembly (Figure S5 is not very informative and adds nothing to the supplement). However, CEGMA predicts 94% complete; how does this compare to other draft genomes? If the genome is as high quality as the authors state, and CEGMA is predicting only 94% complete, what could this mean about *I. nil* gene predictions? For reference, Arabidopsis (TAIR10) is 99.2% complete.

Response: A CEGMA prediction of 90–95% is a usually a good indicator of a high-quality assembly (Kindly refer to <http://www.acgt.me/blog/2014/9/15/understanding-cegma-output-complete-vs-partial>). Analysis of the genomes of tomato, potato, and kiwifruit demonstrated that all these genomes had a CEGMA % completeness lower than 90%.

[Comment of Reviewer #2:]

To this end, the authors predict 41 k genes using only one ab initio gene prediction program (Augustus). While 41 k genes may not be that high, it still is higher than most plant genomes and could reflect erroneous gene prediction.

Response: Ideally, we would have liked to train the gene models using our own dataset. However, owing to limited genomic resources of *I. nil*, we had to use CEGMA gene models to train the dataset. We also tried using BRAKER to train using RNAseq data. However, either way, the gene prediction using tomato-trained models was more accurate than either of them was, when comparing against 189 publicly available CDs sequences. *De novo* based Trinity pipeline was also followed, but resulted in transcripts more than 3 times that of the current prediction. In addition, CEGMA and BUSCO analysis revealed a high completeness ratio. It is possible that there could be errors, but we feel that this is the best possible resource available as of now.

[Comment of Reviewer #2:]

Small RNAs are predicted but not compared to other plant genomes. How do these compare to sweet potato, other plant genomes, and Solanales? Is there anything new or unexpected about the *I.*

nil genome? With the "first pseudo-chromosomal assembly in Convolvulaceae" is there anything special or distinct about this family? Considering the interesting stem behavior and the prominence of the flowers are there anything interesting in regard to the genes that are involved in these processes?

Response: *I. nil* specific gene families were found to be enriched with pollination and reproductive process related gene ontology (GO) terms (Supplementary Table S17), in comparison to the gene families shared with solanaceae members (tomato, potato and capsicum). We plan to perform a comprehensive RNAseq and small RNAseq study in the near future and would like to keep the above recommendations for future work.

[Comment of Reviewer #2:]

I. nil is closely related to the parasitic plant *Cuscuta*, which has significant variation around chloroplast content, size and presence/absence. The authors state that the organellar genomes are deposited at DDBJ under AP017303-AP017304, but they do not provide any analysis in either the main text or supplemental material. Generally PacBio sequence data enables complete assembly of the chloroplast genome and if not the entire mitochondria, it will be in several contigs (usually due to linear or multiple small circles). A statement of the completeness and an analysis of the organellar genomes would add significantly to the genome assembly and analysis.

Response: Thank you for pointing this out. We had included a short section of the organellar analysis in Supplementary material (Supplementary Methods, Supplementary Figs. S4 and S5). We employed both Sanger and PacBio sequencers for the organellar analysis, and the reads produced by the two sequencers were assembled separately. The organellar genomes sequenced using Sanger sequencer were deposited at DDBJ under AP017303-AP017304, and annotations of the genomes were also performed (Supplementary Figs. S4 and S5). The organelle sequences were compared against the PacBio based genome assembly, and we found that several redundant overlapping organelle fragments hitting at same locations. Because of organelle sequence insertions in the nuclear genome as mentioned in the previous report (Heredity 111, 314 (2011)), PacBio contigs seem to have different error profiles even in the overlapping segments of the assembled contigs. Therefore, we just removed the PacBio-based organelle assembly. In terms of completeness, the chloroplast and mitochondrial sequences could be completely reconstructed from just five and three PacBio contigs respectively.

[Comment of Reviewer #2:]

Overall, the authors have presented a very lite analysis of the *I. nil* genome, with a cursory whole genome duplication (WGD) analysis thrown in at the end. However, the main story is about the *Tpn1* transposable elements, which have made morning glory both a fantastic model system as well as ornamentally beautiful. This is where taking the PacBio approach to sequence the genome makes a lot of sense to get the repeat regions; although, the authors don't make a point of this in the text.

Response: Thank you for this comment. We have revised the text accordingly by mentioning that "The 7-kb size selected inserts of the PacBio sequence data was especially helpful in resolving *Tpn1* transposons, whose average length was approximately 7 kb, and the assembly also revealed complex repeats like telomere repeats, rDNA clusters, and centromere repeats. However, a better resolution of such repeats were obtained in *Oropetium thomaenum* genome assembly, possibly owing to the 15-kb lower end insert size selection, explaining the importance of longer read lengths in obtaining near-perfect assemblies."

[Comment of Reviewer #2:]

The high contiguity draft genome assembly is very useful at identifying *Tpn1*-like transposons, and the authors find 322. However, they have to use a cloned version of the *Tpn1* genes from another

line Q1072 to find an almost complete autonomous Tpn1 element. This part of the manuscript is very hard to interpret because it is not well referenced or described. It is not clear why the Tpn1A did not come out of the original search because based on Figure 2 it seems to have the sequence that was used to identify the 322 Tpn1 non-autonomous sequences. Furthermore, they validate and find the complete sequence of Tpn1A with a sequenced BAC (some of Tpn1A is missing in the draft genome assembly); this suggests the genome was not necessary to find Tpn1A and the old fashion way would have worked first and been more effective.

Response: Our explanation about *TpnA1* was insufficient, and may have led to misunderstandings. We have added further explanation on the following points in the Results and Methods sections.

1. We have cloned the putative autonomous element, *TpnA1*, from TKS rather than the line Q1072. The line Q1072 was used only for transposase cDNA cloning. A nucleotide BLAST search against the genome sequence using the cDNA sequences as queries revealed *TpnA1*. We isolated a BAC clone harboring the *TpnA1* sequence from a BAC library made from TKS. The PCR primers used for the BAC library screening were designed based on the genome sequence. The primers were designed in the 3' terminal of *TpnA1* and its 3' flanking sequence. The primer set allowed *TpnA1* specific amplification from the BAC library with not only *TpnA1* but also a number of *TpnA1* related elements. The old fashioned way as described above by the reviewer, as well as the assembled genome sequence were needed for isolation and characterization of *TpnA1*.
2. *TpnA1* as well as *TpnA2* did not come out of the original search, because these transposons were not included in the originally identified 322 *Tpn1* transposons. We used an in-house pipeline for the original search, and the pipeline identifies only *Tpn1* transposons having both 5' and 3' TIR sequences. The 5' terminal of *TpnA1* was not presented in the draft sequence, and the 3' terminal of *TpnA2* is deleted. Therefore, the pipeline failed to identify *TpnA1* and *TpnA2*. We did not use any *Tpn1* transposon sequence for the original search to identify the 322 transposons.
3. For autonomous transposon isolation, we first performed a translated BLAST search against the 322 transposons using TnpA and TnpD sequences from maize and snapdragon as queries. Although we could find that *TpnA3* and *TpnA4* carry TnpD coding sequence, no obvious TnpA coding sequences were found in any *Tpn1* transposons. This is probably due to the fact that only a part of TnpA sequences among *En/Spm* (CACTA) superfamily are conserved (Supplementary Fig. S12c). Actually, however, *TpnA3* and *TpnA4* partially carry TnpA coding sequences (Fig. 2). Therefore, we isolated the cDNA fragments of the transposases from Q1072. The cDNA fragments were necessary not only to isolate *TpnA1* and *TpnA2* but also to predict exons and introns precisely especially in the TnpA coding region.

Owing to updates in the genome sequence, the number of the non-autonomous *Tpn1* transposons increased from 322 to 339 (Supplementary Data S2), and the *TpnA4* sequence slightly changed (Fig. 2). *TpnA1*, *TpnA2* and *TpnA3* sequences in the original and revised genome sequence are identical. No transposons, other than *TpnA1* to *TpnA4* encoding apparently active TnpA and TnpD, were found in the new genome sequence.

[Comment of Reviewer #2:]

Also, Supplementary Data S2 (Predicted Tpn1 transposons) reports the 322 identified non-autonomous Tpn1 transposons. Using this information did the authors find any transposon "scabs" in the TKS reference genome; possibility disrupting coding regions, or islands in the pericentric regions where a history of transposition could be followed?

Response: We tested whether the *Tpn1* transposons disrupt genes in TKS, and the new result was presented in Supplementary Table S14. The *Tpn1* transposons could possibly disrupt at least 29

genes. Further analyses are required to verify that the *Tpn1* transposons actually disrupt any functional genes. On the basis of the positions of *Tpn1* transposons presented in Supplementary Data S2, no islands of *Tpn1* transposons were presented in the genome. The rough positions of the *Tpn1* transposons are also available in Fig. 1.

[Comment of Reviewer #2:]

Finally, the authors clone the CONTRACTED (CT) gene using the genome reference and the classical map position of the mutant. Based on the map position, and the hypothesis that the gene is involved in brassinosteroid biosynthesis they suggest the CT mutation is due to several different types of *Tpn1* insertions into the homologue of the Arabidopsis ROT3 gene. Once again, a genome sequence was not required to identify the CT gene but this is a great use of the draft genome sequence to do reverse genetics. It was not clear from the text or the references whether the mutants analyzed (Table S11) were known previously or whether this was a completely new analysis (sequencing of each line?) presented here. This section is important so the text needs to be clearer as to what was actually done in this study; even the methods section is brief and does not clarify what was actually done here. Moreover, what other genes are candidates in this region (can't tell from Supplementary Data 3

because there is no way to cross reference scaffolds and gene numbers; what is the ROT3 homologue I. nil gene number?) and were any other genes tested for potential *Tpn1* insertions? While, it seems probable that ROT3 is the CT mutant gene, no validation is provided so it is formally possible another linked gene plays a role. If the number of *Tpn1* insertions is ~1000 per genome then there is a chance of another insertion in this region that could explain the phenotype. How can you rule this out? How many non-autonomous *Tpn1* transposons are located in this region in the TKS reference?

[Comment of Reviewer #2:]

Response: Considering the Reviewer's comments, we have revised the explanation of the *CONTRACTED* gene, and have added following 4 points as further explanations.

1. The mutants analyzed had been thought as *ct* mutants from their characteristic phenotypes and allelism tests. The *ct* mutations are classified into *ct* and *ct-w* based on their phenotypes, and the phenotypes of *ct* are slightly severer than those of *ct-w*. The phenotypes in the line used (Supplementary Table S16) are from the database, the National BioResource Project (<http://www.shigen.nig.ac.jp/asagao/>).
2. All mutants were subjected to sequence analysis. Entire genomic *CT* genes in the six lines in the Fig. 3c were analyzed. The lines are Q220 (*ct-1*), Q787 (*ct-1*), Q851 (*ct-2*), Q853 (*ct-2*), Q594 (*ct-w*) and Q848 (*ct-w*), and were used for RT-PCR analysis (Fig. 3c). The sequences of the transposon insertion sites in the other 13 mutant lines were also analyzed (Methods section).
3. The gene ID of the *CT* gene (*ROT3* homologue) is INIL05g09538 (Results section).
4. Among the genes for brassinosteroid synthesis, INIL05g28523 encoding CYP85A2 is located near *A3* and *CT*. The gene is not the *CT* gene, as it is normally expressed in the mutants (Supplementary Fig. S15).

All genes for brassinosteroid synthesis in *I. nil* are listed in Supplementary Fig. S13. The fifth and sixth digits of the gene IDs indicate the chromosome number. INIL05g28523 and INIL05g24193 encoding CYP85A2 and CYP90A1, respectively, were found on chromosome 5 where the *A3* and *CT* genes are located. Of these, INIL05g28523 is located 5.2 - 8.6 cM from the *A3* locus. We have just characterized expression and cDNA sequence of the gene, and found that it is expressed

normally in the mutants. We have presented the new data in Supplementary Fig. S15. Because INIL05g24193 locates far from *A3* locus (55.5-57.8 cM), it is clear that INIL05g24193 is not the *CT* gene.

The 19 tested mutants carry one of the three *ct* alleles (*ct-1*, *ct-2* and *ct-w*) without exception. In contrast, the 24 tested non-dwarf lines do not carry such alleles, without exception. The 43 tested lines are independent lines. Moreover, no *CT* transcripts were accumulated in the mutants (Figure 3c). Thus, we concluded that INIL05g09538 is the *CT* gene of the *ROT3* homologue in the original manuscript. However, we could provide no validation (e.g. transgenic complementation test) as commented by reviewer 2. Therefore, INIL05g09538 is “the most likely (or strong) candidate” for the CONTRACTED gene. We have mentioned this clearly in Introduction and Discussion sections.

No *Tpn1* transposons were located on the scaffold containing the *A3* and *CT* genes. The closest *Tpn1* transposon from *CT* is located 6.6 -6.9 cM from *CT* in TKS. Interestingly, the transposon is identical to *Tpn15*. Since plant transposons preferentially transpose into proximal loci. It is possible that *Tpn15* located in the locus jumped into the *CT* gene in the ancestor of the *ct-w* lines.

[Comment of Reviewer #2:]

Minor suggestions:

Add genome size to the abstract and introduction.

Response: We have added the genome size to both abstract and introduction, as suggested.

Claims of being the "first" are not necessary if the results are significant; remove from the abstract and introduction.

Response: We have removed the word “first”, as suggested.

Page 7. "In case of mis-assemblies at contig level," should read "In the case of mis-assemblies at the contig level."

Response: We have corrected the above sentence accordingly.

[Comment of Reviewer #2:]

Page 7. "...achieved utilizing traditional Sanger sequencing data." Needs a reference.

Response: We have placed a reference citing first 50 sequenced plant genomes.

[Comment of Reviewer #2:]

Page 9. Reporting the 3:2 copia to gypsy ratio as 3:2 is misleading since the 22% represents uncharacterized copia and gypsy elements specific to *I. nil*, which could be updated by annotating the unclassified.

Response: Thank you for pointing this out. It was indeed misleading, and we apologize for that. We extracted unknown elements and classified them by aligning them against RepBase sequences. We found 12.9 % copia and 14.4 % gypsy elements in total.

[Comment of Reviewer #2:]

Page 9. "...TpnA and TnpD transposase coding sequence." Needs a reference.

Response: We have replaced this sentence and added a reference as follows:

“It could be expected that the autonomous *Tpn1* family transposons carry both the TnpA and TnpD transposase coding sequences such as *En/Spm* and related autonomous transposons¹¹.”

[Comment of Reviewer #2:]

Differential expression is presented in Supplementary Data S4 (Scaled FPKM values of clustered DEGs); are there replicates for that data? If not then what does this data really mean?

Response: There were no replicates and hence, we have omitted the RNAseq results. In the revision,

we have used RNAseq data only for gene prediction purposes.

[Comment of Reviewer #2:]

Page 14. *Vigna angularis* and *Oropetium thomaeum* are mentioned in the discussion but they do not appear in Table S12.

Response: Supplementary Table S12 corresponds to articles where PacBio was used for scaffolding the Illumina only contigs. Whereas, *Vigna angularis* and *Oropetium thomaeum* use PacBio reads for the initial assembly and hence were not related to that table (Supplementary Table S18). However, we have included the results as per the suggestion.

We thank the reviewer once again for the constructive comments.

[Comments of Reviewer #3:]

[Comment of Reviewer #3:]

I have reviewed the manuscript, "Genome sequence analysis of the Japanese morning glory, *Ipomoea nil*", by Hoshino et al. This is a basic genome sequencing paper with some relevant analysis about transposable elements content and about comparative genomic analyses with other sequenced dicot species. This paper was a pleasure to read. It is well written, and I appreciate that a great deal. The authors used PacBio sequences to generate their genome assembly. In fact, they also performed Illumina sequencing, but the resulting assembly was not quite as good as the PacBio assembly. The Illumina sequence was used for assembling scaffolds from the PacBio unitigs and for gap filling. This procedure was explained well, and the Illumina vs PacBio comparison is very useful. If you have a budget that will allow PacBio sequencing, an impressive genome assembly can be obtained even for a medium sized genome. Otherwise, you can have a perfectly functional genome assembly for a more affordable cost.

Response: We thank the reviewer for the positive and encouraging comments.

[Comment of Reviewer #3:]

I do have issues/concerns.

The authors used CEGMA to assess the completeness of their genome assembly. CEGMA is no longer the standard for this assessment. BUSCO is the new tool that should be used for this purpose. Even the author of CEGMA has indicated that BUSCO should be the standard tool for assessing genome completeness. In fact, it is acceptable to also report CEGMA results, but as it is relatively easy to run, the authors should use BUSCO on their genome assembly. The results from BUSCO will not show 90+% completeness. Even rice and arabidopsis only get BUSCO scores in the 80's. Even without BUSCO results, the other controls that the authors have included indicate that the assembly is very good.

Response: As suggested, we have added the BUSCO results to the revision. There is an early release of BUSCO aimed at plant genomes, which gives Arabidopsis a score of 97% completeness. For *I. nil*, the score was 95%.

[Comment of Reviewer #3:]

I am really disappointed that the authors used the Augustus tomato HMM for their gene prediction. This is very bad practice. In the article that describes the SNAP gene annotation program, Ian Korf demonstrates why using an HMM from one species to predict genes in a second species is a bad idea. The utility of a SNAP HMM to predict genes in a new genome is not correlated with phylogenetic similarity between the species used to train the HMM and the target species. The utility of a SNAP HMM to predict genes in a new genome is correlated with the GC content of the genes in the two species. I am very close to nixing this submission over this issue. Gene prediction HMMs do not accurately predict genes with GC content that is wildly different than the GC content

of the genes that had been used to train the HMM. Predictions will be made, but they are often not accurate. Augustus should have been trained with example genes from *Ipomoea nil*. The process is obscure but not difficult. The Augustus developers will even help with the training. I would like the authors to calculate the GC content of the CDSes from *Ipomoea nil* ESTs and/or fl-cDNAs. A couple hundred of these sequences would be sufficient. The GC content of the predicted CDSes from the tomato genome should also be calculated. These two GC content values should be very similar (within about 5%) to be able to argue that using the August tomato HMM was acceptable. I think that a difference in these GC values of more than 10% would cause me to be concerned about the gene annotation. Another number that would help to indicate that the *Ipomoea nil* genes were well annotated would be the total number of genes that had been in an OrthoMCL ortholog group with at least one other species. I don't think that this number was presented.

Response: As suggested by the reviewer, we trained AUGUSTUS using CEGMA predictions due to a scarcity of complete CDs of *I. nil*. Independently, we also used BRAKER software, which used RNAseq alignments to train the *I. nil* genome. When we predicted genes separately in those 2 trained models, we arrived at more than 55,000 genes in both the cases. However, gene predictions using tomato based gene models resulted in approximately 42,783 gene models, probably owing to tomato's large training data size. To crosscheck, we aligned the 189 complete CDs sequences of *I. nil* from NCBI against the genome. Manual inspections of the alignments revealed that 116 out of 189 CDs of *I. nil* were predicted to be complete in the tomato based predictions, compared to 61 out of 189 CDs in the self-trained gene models. In addition, we calculated the GC% of tomato cDNAs (40.61%) and NCBI cDNA of *I. nil* (44.57%). Because the GC content range is within 5% difference and the tomato-trained gene models are more in line with the actual CDs, we decided to use the gene prediction models based on tomato trained models.

[Comment of Reviewer #3:]

The tRNA results were very interesting. However, I have been taught that if a result is not interesting enough to mention in the Discussion section, then it does not need to appear in the Results section. I would like to see some analysis of this tRNA finding in the Discussion section. Are there some other analyses beyond the Infernal analysis that could be run to support the identification of so many tRNAs? Wow. This is out of control if true.

Response: Yes, the results do seem interesting. However, at the moment, we are not aware of detailed tRNA analysis tools. We realize that it is probably out of scope to cover in this manuscript. For now, we have omitted these results. We will come up with collaborations in the future for an in-depth study of the tRNAs.

[Comment of Reviewer #3:]

In the discussion, the authors state that *Tpn1* transposons are the major mutagen in *Ipomoea nil*. That is a pretty emphatic statement that I don't think has been sufficiently supported. Are *Tpn1* transposons more active than GC-biased gene conversion or 5-methylcytosine to thymine conversion? I cannot let this one slide. Something has to change in the text.

Response: Thank you for pointing this out. *Tpn1* transposons are active only in the mutant cultivars that have been isolated since the early 19th century in Japan. Recently, several of our studies revealed that the transposons in the *Tpn1* family cause a large proportion of the mutations. Among the 19 mutations that have been reported, 10 mutations are insertion of *Tpn1* family elements, and 3 are 4-bp insertions that are putative footprints of *Tpn1* family elements. Only 2 of 19 mutations (C to T and A to T conversions in *mg* and *star* mutations, respectively) are single base substitutions. This indicates that the *Tpn1* transposons are more active than GC-biased gene conversion or 5-methylcytosine to thymine conversion in the mutant cultivars. We have revised the text as follows: "These features should be the basis for *Tpn1* transposons to act as the major mutagen in the mutant cultivars of *I. nil*."

We thank the reviewer once again for the constructive comments.

Dear Reviewers,

Please find below the responses to the reviewer's comments.

[Comment of Reviewer #2:]

As a point of clarification, this reviewer wanted to have access to the genome to validate quality and claims. Since the genome was not available for scrutiny by the reviewers, the claims concerning the genome will have to be taken at face value.

Response: The access to our data at DDBJ has been made open to public from August 31, 2016.

[Comment of Reviewer #2:]

One small sticking point brought up by one of the other reviewers is the gene prediction based on tomato gene models. The authors make a good case for using the tomato models in their rebuttal. However, based on the information in the introduction, there seems to be ample information to make *I. nil* models:

Lines 103-105: "62,300 expressed sequence tags (ESTs) deposited to the DDBJ/EMBL/NCBI databases, Simple Sequence Repeat (SSR) markers and a recent large scale transcriptome assembly".

And

Lines 182-184: "Comparisons against 93,691 ESTs showed that 99.11 % of them were aligned, with 97.40 % of the ESTs having at least 90 % of their lengths covered in the alignments.

A path forward would be to make it very clear that tomato models were used by adding a statement in lines 262-263 that is similar to what is in the methods section:

Line 510: "Gene models were predicted using Augustus v3.2.229 with tomato as the reference species,..."

Response: We have modified the text accordingly as "A total of 42,783 gene models were predicted along with 45,365 transcripts, with tomato as the reference species, using Augustus".

[Comment of Reviewer #2:]

Also, the rebuttal explanation (or a concise version of it) would add value and clarity in the methods section.

Response: The following text is included: "Because of the scarcity of complete CDs of *I. nil* in public databases, independently, Augustus was also used to predict gene models, after training using CEGMA predicted genes, and the procedure resulted in more than 55,000 gene models. The 189 complete CDs sequences already available in NCBI were downloaded and compared against the predicted gene models using BLAT. Tomato based gene models showed that 116 out of 189 CDs were perfectly complete, whereas CEGMA trained gene models showed that only 61 out of 189 CDs were complete and hence, the tomato based gene predictions were used for further analysis."

[Comment of Reviewer #2:]

Lines 101-103

It would flow better if these two sentences were edited to make it clear that there are 15 chromosomes and 10 of them have mutants mapped to them.

I. nil has 15 pairs of chromosomes ($2n = 30$)¹⁵. A total of 219 genetic loci of *I. nil* had been analyzed by 1956. Among them, 71 loci were mapped to one of the 10 linkage groups (LGs).

Response: The text is edited as, "However, the original classical map from 1956 contained

only ten linkage groups, as a result of mapping 71 genetic loci out of 219 analyzed loci to one of the ten linkage groups".

[Comment of Reviewer #2:]

Lines 108-111

Should read "average scaffold length."

The genome of a closely related species of a wild sweet potato, *I. trifida*, was recently sequenced and published, in which they reported genome sequences of two *I. trifida* lines analyzed using Illumina HiSeq platform, with an average length of 6.6 kb (N50 = 43 kb) and 3.9 kb (N50 = 36 111 kb), respectively.

Response: The average length is edited to average scaffold lengths.

[Comment of Reviewer #2:]

Lines 302-303

214 what? Restructure the sentence to make it clear as to what 214 refers to.

A total of 1,353 single copy orthologs corresponding to the seven species were extracted from the clusters and were filtered to 214.

Response: The text is edited as 214 single copy orthologs.

[Comment of Reviewer #2:]

Lines 336-337

Highly contiguous would sound better than long because a long genome assembly does not make sense.

"variety of species. The current study has utilized nearly the complete potential of recent sequencing tools and has culminated in a long, high quality genome assembly of *I. nil*."

Response: Long is edited to highly contiguous.

[Comment of Reviewer #2:]

Line 784

Figure legend 4, define the color and lines within the text of the legend.

Response: The colors of the lines are stated in the figure legend.

All the above changes are tracked using MS word.

We hope that we have addressed all the changes suggested by the reviewer.

Sincerely,

Yasubumi Sakakibara

Professor, Department of Biosciences and Informatics, Keio University, Japan.